# Graded titin cleavage progressively reduces tension and uncovers the source of A-band stability in contracting muscle

Yong Li[†], Anthony L Hessel[†], Andreas Unger[†], David Ing, Jannik Recker, Franziska Koser, Johanna K Freundt, Wolfgang A Linke*

Institute of Physiology II, University of Muenster, Muenster, Germany

**Abstract** The giant muscle protein titin is a major contributor to passive force; however, its role in active force generation is unresolved. Here, we use a novel titin-cleavage (TC) mouse model that allows specific and rapid cutting of elastic titin to quantify how titin-based forces define myocyte ultrastructure and mechanics. We show that under mechanical strain, as TC doubles from heterozygous to homozygous TC muscles, Z-disks become increasingly out of register while passive and active forces are reduced. Interactions of elastic titin with sarcomeric actin filaments are revealed. Strikingly, when titin-cleaved muscles contract, myosin-containing A-bands become split and adjacent myosin filaments move in opposite directions while also shedding myosins. This establishes intact titin filaments as critical force-transmission networks, buffering the forces observed by myosin filaments during contraction. To perform this function, elastic titin must change stiffness or extensible length, unveiling its fundamental role as an activation-dependent spring in contracting muscle.

**\*For correspondence:**
wlinke@uni-muenster.de

[†]These authors contributed equally to this work

**Competing interests:** The authors declare that no competing interests exist.

## Introduction

The theories of muscle contraction were originally developed under the assumption that active force of a muscle cell's contractile unit, the sarcomere, was entirely generated by the interactions of thick and thin filaments (*Gordon et al., 1966*; *Huxley, 1957*). Many years later, a third sarcomeric filament called titin (also known as connectin) was discovered (*Maruyama, 1976*), and research since then has demonstrated that titin is a critical component of sarcomeric function (*Linke, 2018*; *Nishikawa, 2020*). Titin is the largest known protein (*Bang et al., 2001*) and spans half of the sarcomere from the Z-disk to the M-band. The titin gene is truncated in a substantial proportion of the general population, and these truncations are frequently associated with cardiomyopathy and occasionally with skeletal myopathies (*Linke, 2018*). The I-band portion of titin links the thick and thin filaments as an extensible and viscoelastic spring. While the I-band titin spring produces most, if not all, of the passive force in sarcomeres, the role of titin during active contraction remains undefined, but yet seems critical because the (unspecific) destruction of titin coincides with passive *and* active tension loss (*Higuchi, 1992*; *Horowits et al., 1986*). In agreement, changes to I-band titin affect active force production but are not easily explained under the current paradigm of muscle contraction (*Linke, 2018*; *Nishikawa, 2020*). For example, skeletal muscles with genetically modified I-band titin show altered mechanical properties, including muscle stiffness and force, length-dependent activation, and mechanosignaling (*Brynnel et al., 2018*; *Buck et al., 2014*; *Mateja et al., 2013*). A relationship between titin-based and actomyosin-based forces has been suggested to optimize the work generated by muscle contraction (*Rivas-Pardo et al., 2016*). However, direct empirical evidence into titin-based regulation of active contraction is limited.

To study titin's role in passive and active force production, the most straightforward experimental strategy is to compare the mechanical properties of muscle before and after the functional removal

of titin. Classic techniques include genetically impairing titin synthesis (*Radke et al., 2019*; *Swist et al., 2020*), chemical degradation of titin via trypsin digestion (*Higuchi, 1992*), or protein destruction via target ionizing radiation (*Horowits et al., 1986*). However, almost all animal models with a mutational change in titin present a phenotype of muscle dystrophy and wasting (e.g. *Radke et al., 2019*; *Swist et al., 2020*), and all previous methods to degrade titin also degrade other proteins (*Horowits, 1999*). Therefore, it is difficult to identify data trends that are caused directly by the changes in titin or indirectly by the consequences of the ensuing 'disease' state.

To mitigate the problems outlined above, a new mouse model was generated with a cloned-in HaloTag-TEV cassette inserted into I-band titin close to the A-band (*Rivas-Pardo et al., 2020*). We refer to this mouse as the titin cleavage (TC) model. The tobacco etch virus (TEV) protease-recognition site is specifically cleaved by the TEV protease, while the HaloTag domain allows for easy protein labeling, useful, for example, for the assessment of titin cleaving. The insertion itself does not affect mouse development, muscle structure, or performance (*Rivas-Pardo et al., 2020*). Our experimental strategy is to use TEV protease to cleave I-band titin and terminate its functionality in a targeted and controllable fashion. This abrupt change to otherwise-normal sarcomeres allows us to precisely track the immediate changes in sarcomere structure and function with titin loss, and study as purely a titin-based effect as possible.

The purpose of this study is to use the TC model to define the role of titin-based forces in skeletal muscle contraction and sarcomere integrity. We achieve this by cleaving 0%, ~50%, and 100% I-band titin molecules in permeabilized wild-type (Wt), heterozygous (Het), and homozygous (Hom) TC fibers, respectively. We find that passive and active forces become progressively reduced as TC doubles from Het to Hom TC muscles, and we observe a striking disruption of Z-disk alignment. Remarkably, we uncover a previously unknown function of titin to support M-band structures by partially bearing the opposing forces that develop between adjacent thick filaments during contraction. Based on the established properties of titin elasticity and force, we find that titin's mechanical buffering role requires a substantial increase in titin stiffness during contraction or a shortening of the extensible length of elastic titin, which we propose to take place by an activation-dependent enhancement of titin-actin binding.

## Results

### Different TC muscle genotypes allow graded TC

Initially, we quantified the proportion of titin molecules containing the HaloTag-TEV cassette by antibody labeling the HaloTag in psoas skeletal muscles of all three genotypes (Wt, Het, Hom) (*Figure 1A*). Both confocal immunofluorescence (IF) and immunoelectron microscopy (IEM) indicated the absence of HaloTag labeling in Wt TC fibers, whereas in Het and Hom TC fibers, HaloTag signals localized to their expected I-band (near I/A-band junction) position in the sarcomeres (*Figure 1B*). On IF images, a signature doublet pattern of HaloTag labeling appeared. We used ACTN2, the Z-disk marker of α-actinin, as an internal normalization signal for HaloTag IF intensity. Het produced 57.3% of normalized HaloTag IF staining intensity compared to Hom TC fibers (Het: 32.08 ± 0.90 arbitrary units (a.u.; mean ± SEM, Hom: 55.95 ± 0.64 a.u.; *Figure 1C*). When comparing the label abundance of HaloTag antibody-conjugated nanogold particles on IEM images between genotypes, Het contained 36.8% of Hom (Het: 34.42 ± 2.42 a.u., Hom: 93.60 ± 1.98 a.u., *Figure 1D*). We continued this assessment by treating permeabilized (skinned) TC skeletal muscle fibers with TEV protease for 6 hr, followed by titin analysis on Coomassie-stained protein gels (*Figure 1E*). Compared to a single N2A titin band (~3.7 MDa) in control samples (including treated and untreated Wt), after TEV protease treatment, two smaller bands were clearly visible in Het and Hom TC muscles at the expected sizes of ~2.3 MDa for A-M-band titin and ~1.4 MDa for Z-I-band titin, indicating specific titin cleavage. The ~2.3 MDa titin cleavage product was slightly smaller than a doublet running at ~2.4 MDa, which contains proteolytic titin fragment T2 and the alternative titin isoform, Cronos (*Swist et al., 2020*; *Figure 1E*). Densitometric analysis of intact and A-M-band titin intensities confirmed full cleavage in Hom fibers and revealed 43.12 ± 0.85% cleaved titins in Het (*Figure 1E*). However, the 'gold standard' for protein quantification is western blotting, which we performed with anti-HaloTag antibodies following titin gel electrophoresis. We found that the proportion of mutant titins in Het was 49.86 ± 1.24% (n = 29) of Hom (100.00 ± 5.51%; n = 13) (*Figure 1F*). These data

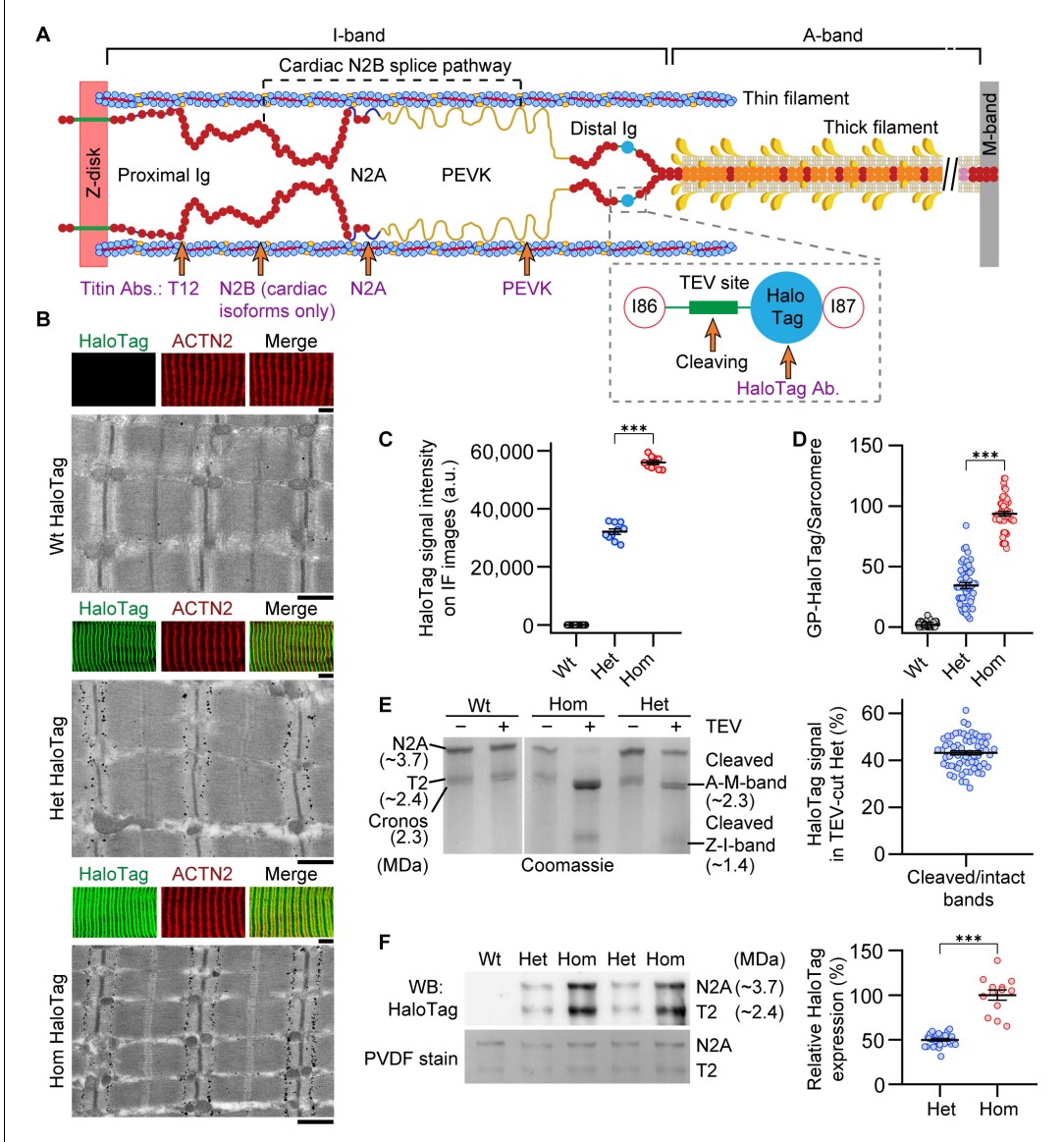

**Figure 1.** Mutant titin expression in skeletal muscles of different genotypes. (**A**) Schematic of titin cleavage (TC) mouse half-sarcomere. Distal I-band titin holds a tobacco etch virus (TEV) protease-recognition site and a HaloTag. The epitope positions of titin and HaloTag antibodies are indicated. (**B**) Correlative immunofluorescence (IF) and immunogold electron microscopy of wild-type (Wt), heterozygous (Het), and homozygous (Hom) psoas muscle. Representative IF images of fibers (colored panels) labeled with HaloTag antibody (green) and counterstained for α-actinin (ACTN2, red), and immunoelectron micrograph showing HaloTag labeling. Scale bars, 5 μm (IF); 1 μm (IEM). (**C**) Quantification of HaloTag-expression level using IF intensities (HaloTag signals normalized to ACTN2 signals; n = 10 fiber averages; sarcomere length [SL] range, 2.7–2.9 μm). (**D**) Gold particle (GP) count per sarcomere, from HaloTag-stained IEM images (n = 50 sarcomeres/group; SL range, 2.7–2.9 μm). (**E**) Coomassie-stained, agarose-strengthened 1.8% sodium dodecyl sulfate–polyacrylamide gel electrophoresis (SDS–PAGE), titin gel analysis in different genotypes. Left: Representative gel with bands labeled. Right: Results of densitometric quantification of cleaved titin (A-M-band) intensity, compared to non-cleaved titin in Het (n = 65 samples; n = 5 different mice/group). (**F**) Western blot analysis of mutant titin (N2A) expression in different genotypes. Left: Representative immunoblot using anti-HaloTag (top) and Coomassie-stained PVDF membrane indicating protein load (bottom). Right: Densitometric results for Het vs. Hom anti-HaloTag signal intensities (n = 29 Het, 13 Hom). Stats: ANOVA with Tukey's HSD post hoc procedure or Student's t-test. The online version of this article includes the following source data for figure 1:

**Source data 1.** Data, Stats, and uncropped gel/blot figures for *Figure 1*.

show that on average, approximately 50% of HaloTag-TEV titins are present in Het fibers and that these molecules are effectively cleaved by TEV protease.

## Passive longitudinal force reduction scale with the amount of TC

To estimate the contribution of titin to passive force and stiffness, measured in a 'relaxing' buffer, we compared stretch-dependent forces from ramp-hold experiments (*Linke et al., 1998b*) before and after titin cleavage in Wt, Het, and Hom permeabilized psoas fiber bundles (*Figure 2*). Average sarcomere length (SL) was measured using laser diffraction. The fibers were stretched from 2.2 μm SL (slack) to 3.4 μm SL at ~0.2 μm SL s$^{-1}$ in a stepwise fashion, followed by a release back to slack, and the process repeated.

Next, TEV protease was added, and a waiting time of 10 min was observed, sufficient to cleave ~50% and ~100% TC titin in Het and Hom fibers, respectively (see below). Periodic ramp-hold cycles then continued for an additional 30 min, and the final force trace was compared to the force trace immediately before the addition of TEV protease. Titin cleavage readily resulted in a length-dependent decrease of passive tension in Het and Hom TC fibers, but not Wt fibers (*Figure 2A*). Fiber force was separated into velocity-insensitive (elastic; *Figure 2B,C*) and velocity-sensitive (viscous; *Figure 2D,E*) components as titin has viscoelastic properties. Both components were reduced in the mutant genotypes after titin cleavage, more so in Hom than in Het (*Figure 2B,D*). The relative force decreases in Hom vs. Wt typically reached ~50–55% at SLs between 2.6 and 3.2 μm. Next, we searched for a correlation between decreasing passive force components and mean proportion of intact titins after TEV-protease treatment at each SL (*Figure 2C,E*). We found that negative correlations were strongest ($R^2 > 0.5$) at moderate or long SLs, with the slope of the regression becoming steeper at the longer SLs. At the shortest SLs, we found weak correlations ($R^2 < 0.5$) as forces were always close to 0, which is in line with the relatively small titin- or fiber-based passive forces at these lengths, as well as the non-linear nature of the titin force–SL relationship (*Prado et al., 2005*). Hom and Het data was fit well by a linear curve, indicating that the proportion of titin cleavage was nearly the only factor affecting force in both genotypes.

Note that in glycerol-skinned fiber preparations, some but not all extra-myofibril components to passive force are removed; thus, non-titin contributions to passive force will be realized in passive force traces. Indeed, non-titin force components clearly appeared in treated Hom fibers, where nearly 50% of passive elastic forces remained between 2.4 and 3.2 μm SL and >50% remained at 3.4 μm SL. These residual forces after titin cleavage in Hom skinned fibers are presumably due to strained intracellular (e.g. microtubules; *Kerr et al., 2015*) and/or extramyofibrillar structures (*Prado et al., 2005*). Taken together, these results provide a precise quantification of titin's contribution to permeabilized fiber viscoelasticity under relaxing conditions and substantiate the function of titin as a main determinant of passive, longitudinal cellular forces.

## Radial compressive forces depend less on intact titin stiffness than tensile stretch forces

An often-neglected passive mechanical property of titin is its contribution to transverse stiffness (*Li et al., 2016*). As I-band titin is cleaved, this radial stiffness component will decrease, which is predicted to also decrease the magnitude of compressive forces the myocyte can withstand, and increase its indentability. We evaluated titin cleavage-induced changes to transverse (compressive) stiffness by nanoindentation using the atomic force microscope (AFM). We measured radial indentation depth and stiffness (Young's modulus; *Li et al., 2016*) before and after TEV treatment of skinned, single, Wt and Hom psoas fibers (*Figure 3A*). We found that titin loss in Hom fibers increased the indentation depth at 3 nN indentation force ~33% from 295.90 ± 5.97 nm to 392.70 ± 11.60 nm, and the Young's modulus decreased ~33% from 4.81 ± 0.15 kPa to 3.24 ± 0.15 kPa (mean ± SEM; n = 34/38 recordings for Hom/Wt) (*Figure 3B*). Thus, titin-based transverse stiffness is an important component of cellular transverse stiffness in skinned skeletal fibers. However, radial compressive forces are relatively less dependent on intact titin stiffness than longitudinal stretch forces.

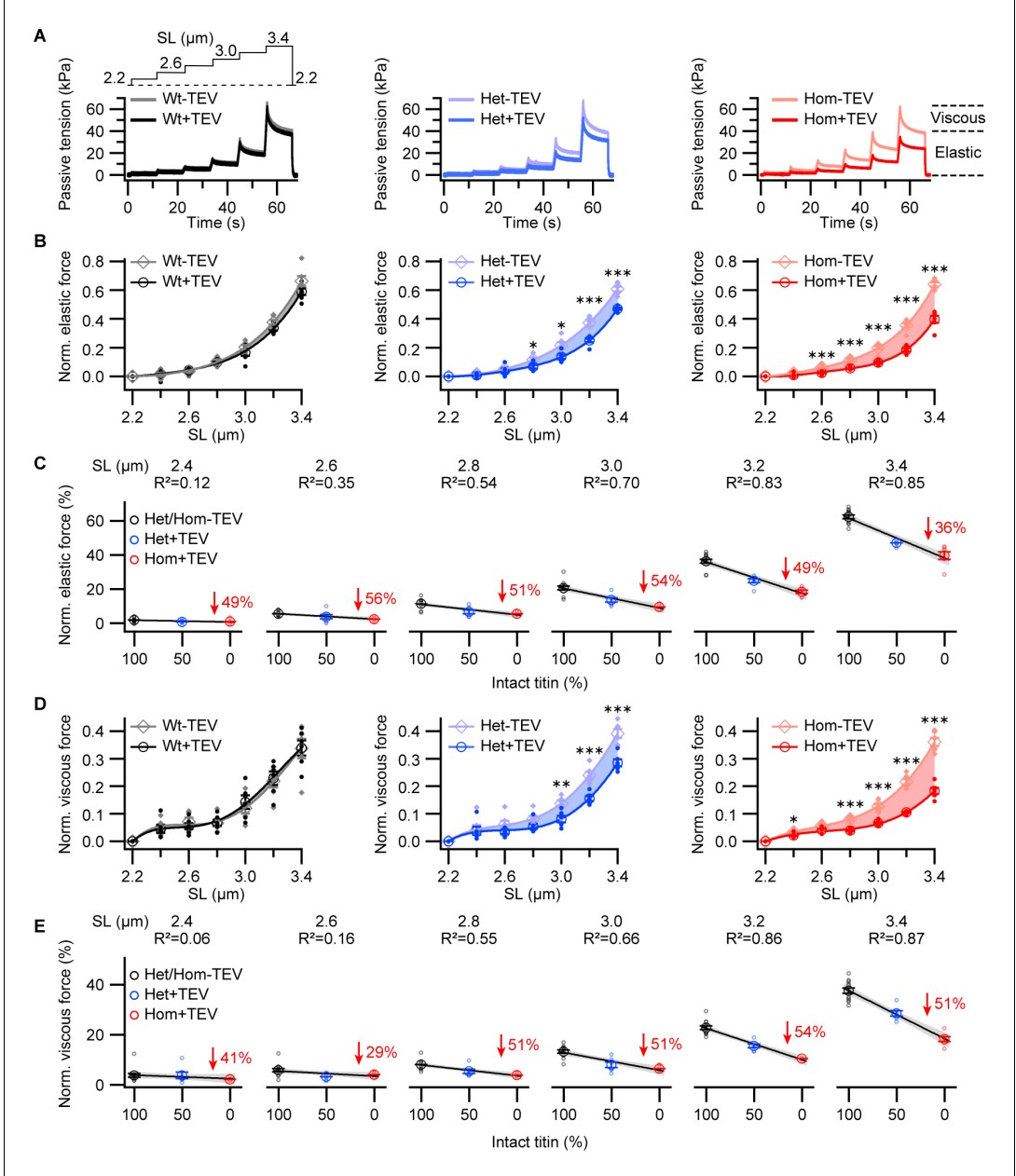

**Figure 2.** Passive forces of permeabilized titin cleavage (TC) fibers before and after titin cleavage. (**A**) Representative tension-time traces from passive ramp experiments, for permeabilized Wt, Het, and Hom fiber bundles. Fibers were stretched stepwise (protocol on top, left), and passive force was recorded before and after 30 min of TEV-protease treatment. Data were normalized to maximum peak force before TEV treatment for analysis. (**B**) Average elastic and (**D**) viscous tension reduction with TEV-protease treatment, per genotype (n = 6 fiber bundles/group). Significant differences after TEV protease treatment are identified (asterisks). We further conducted a correlation analysis between passive force and intact titin content as a function of SL for both elastic (**C**) and viscous (**E**) components. A linear model was fit to force vs. intact titin data at each SL for Wt (black), Het (blue), and Hom (red) trials. Passive force drop between control and TEV-protease treatment for Hom shown in red. Gray shade around linear regressions indicates the 95% confidence interval. At short SLs, where titin-based forces are very small, no relationship, or a weak negative relationship, exists ($R^2 < 0.5$), which becomes stronger at moderate and long SLs ($R^2 > 0.5$). Stats: ANOVA with Tukey's HSD post hoc procedure.

The online version of this article includes the following source data for figure 2:

**Source data 1.** Data and Stats for *Figure 2*.

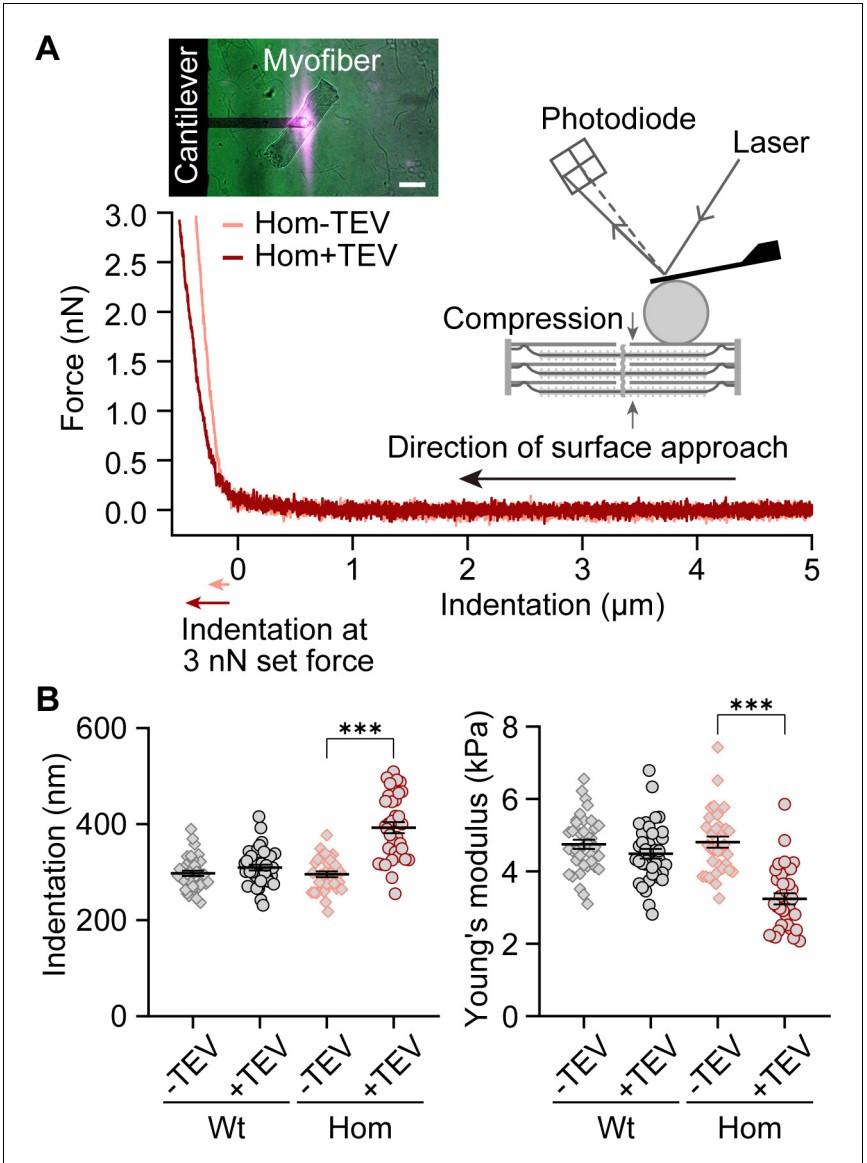

**Figure 3.** Transverse fiber stiffness upon titin cleavage by atomic force microscope (AFM) nanoindentation. (**A**) Representative single force-indentation curves obtained from relaxed, permeabilized Hom psoas fibers before (orange) and after (red) TEV protease treatment. The principle of AFM measurement is depicted in the cartoon. Inset on upper left: image of a fiber during AFM experiments. Scale bar, 100 µm. (**B**) Indentation depth at 3 nN (left) and Young's modulus (right) for Wt and Hom fibers before and after TEV protease treatment (Wt: n = 38 recordings, Hom: n = 34 recordings; six different fibers per group were analyzed). Stats: ANOVA with Tukey's HSD post hoc procedure.

The online version of this article includes the following source data for figure 3:

**Source data 1.** Data and Stats for *Figure 3B*.

## Titin cleavage causes Z-disk misalignment under stretch

We continued to characterize the consequences of specific titin cleavage on fiber properties, this time inducing the cleavage in a highly extended state. Permeabilized fiber bundles were held at 3.3 µm SL and force recorded for 30 min after the addition of TEV protease (*Figure 4A*, left and middle). We found that, compared to control trials, TEV-protease rapidly decreased passive force for the first 10 min after its addition to Het or Hom TC fibers (*Figure 4A*, middle). Thereafter, the kinetics were similar to Wt fibers, indicating that most cleavable titins were severed after ~10 min of treatment,

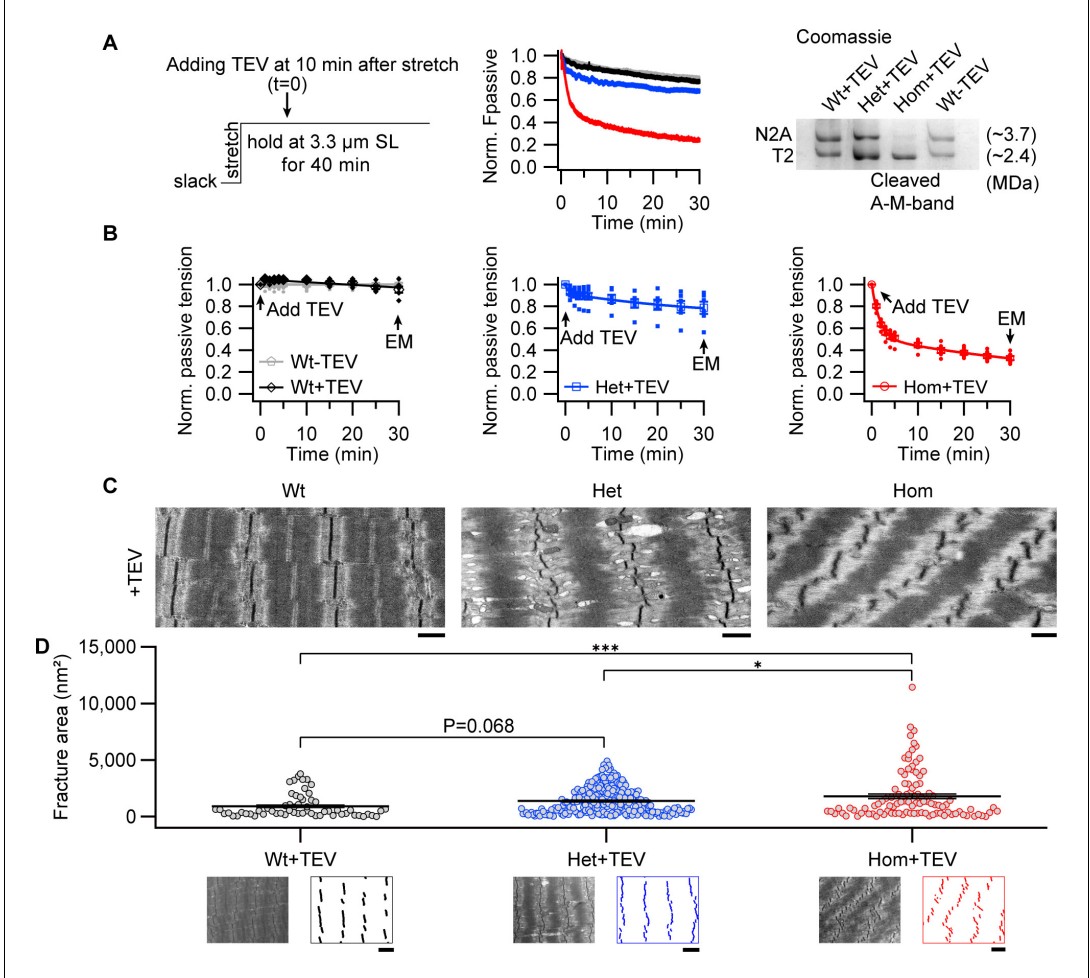

**Figure 4.** Z-disks become increasingly misaligned with progressive titin cleavage in the stretched state. (**A**) Time-resolved passive tension decreases with or without TEV-protease treatment, at 3.3 µm SL; experimental protocol (left) and representative tension-time traces (middle). Raw data was normalized to the averaged time-matched tension without TEV protease treatment. Following mechanical measurements, a subset of fibers was analyzed by titin protein-gel electrophoresis to confirm expected cleavage (Coomassie-stained example, right). (**B**) Averaged values at 0, 1, 2, 3, 4, 5, 10, 15, 20, 25, and 30 min after TEV protease treatment or no treatment for control (n = 6 WT, 6 Het, 6 Hom animals). (**C**) Representative electron micrographs of relaxed Wt, Het, and Hom fibers after treatment with TEV protease while being held passively for 30 min at the stretched length. Scale bars, 1 µm. (**D**) Z-disk order of TEV protease treated fibers after hold protocol, quantified by fracture area (see Materials and methods; Wt+TEV: n = 60, Het+TEV: n = 239, Hom+TEV: n = 104). The images below the graph show representative electron micrographs and processed images used to calculate fracture area, for each genotype, after TEV treatment. Scale bars, 1.5 µm. Average length of sarcomeres on images, 3.3 µm (range, 3.0–3.6 µm). Stats: ANOVA with Tukey's HSD post hoc procedure.

The online version of this article includes the following source data for figure 4:

**Source data 1.** Data, Stats and uncropped gel figure for *Figure 4*.

slightly faster than previously reported by us (*Rivas-Pardo et al., 2020*), due probably to different activities of the TEV-protease batches used. Protein gels confirmed ~50% cleavage for Het TC samples and ~100% cleavage for Hom (*Figure 4A*, right). In Het and Hom (but not Wt) TC samples, TEV-protease treatment removed titin-based forces accounting for 21.42 ± 4.26% and 67.44 ± 1.46%, respectively, of total passive force (*Figure 4B*). The decrease in Het was slightly lower, and that in Hom somewhat higher, than in the stepwise-stretch protocols (*Figure 2*), but overall, the results confirmed the crucial role of titin for permeabilized fiber stiffness.

Next, we used these stretched fibers on which passive forces had been recorded, to investigate the impact of titin cleavage on muscle ultrastructure (*Figure 4C,D*). The relaxed, permeabilized fibers (Wt, Het, and Hom TC fibers) were fixed at 3.3 µm SL immediately after the experiment and prepared for transmission electron microscopy (EM, *Figure 4C*). Wt fibers treated with TEV protease

appeared structurally normal, whereas TEV protease-treated Het and Hom TC fibers revealed A-band disorder (streaming) and Z-disks that lost their typical linear appearance. We quantified this Z-disk 'waviness' in EM images by comparing Wt with Het and Hom fibers fixed at average SL 3.3 μm. To this end, we developed custom software (see Materials and methods) that extracts Z-disk pattern data from EM images and calculates a variable that quantifies the level of Z-disk disorder, which we term 'fracture area' (*Figure 4D*). According to this analysis, when compared to the WT fibers, TEV protease-treated Het fibers had a modest increase in fracture area, whereas Hom fibers showed substantial Z-disk disorder, with large variability (*Figure 4D*). Taken together, the observed decrease in Z-disk linearity with progressive titin cleavage demonstrates the importance of titin-based forces in balancing out the I-bands on opposite sides of a Z-disk, which is an interesting, previously unrecognized, property of titin.

## After cleavage, recoiling titins stick to the thin filaments

We expected the elastic I-band titin to recoil toward the Z-disk after TEV-protease cleavage. To characterize this recoil, a subset of samples from the passive stretch-hold experiments at 3.3 μm average SL, where titin-based recoil forces would be large, were prepared for immuno-EM and were labeled with I-band titin antibodies against the near-Z-disk (anti-T12), central I-band (anti-N2A), and near A-band (anti-proline-glutamate-valine-lysine-rich (PEVK))regions (*Figure 5A*; for antibody-epitope position, see *Figure 1A*). For quantification, the nearest Z-disk-to-epitope distance was measured in TEV-treated fibers, with decreasing values indicating recoil toward the Z-disk (*Figure 5B*). As a secondary assessment of titin recoil, we also immunolabeled these titin regions for confocal microscopy and compared to the Z-disk stain of ACTN2 (*Figure 5—figure supplement 1*). If the labeling patterns of ACTN2 and titin merged, then this would indicate full titin recoil; we observed this phenomenon as expected (*Figure 5—figure supplement 1*, 'merged' images). On immunoelectron micrographs, the proximal T12 labels (nanogold particles) remained stationary upon titin cleavage, relative to the Z-disk, whereas the two other more distal antibody labels were partially translocated toward the Z-disk, indicating titin recoil (*Figure 5B*). In Het fibers, a bimodal distribution appeared, with some epitopes remaining at the normal position and others recoiling partially or fully back to the Z-disk. Interestingly, even in Hom fibers, full titin recoil was not achieved by many titin molecules, as reflected by a high variability in recoil distance for N2A and PEVK epitopes (*Figure 5B*).

Full titin recoil was not achieved, likely because titin stuck to the surrounding proteins of the thin filament, a known interaction point in in vitro preparations (*Kulke et al., 2001*). The most straightforward way to test this hypothesis was to remove actin and assess if cleaved titins recoiled further. Specific removal of thin filaments was possible by treatment with a $Ca^{2+}$-independent gelsolin fragment, although this approach works well only in cardiac muscle (*Linke et al., 1997*), as confirmed for this study using permeabilized Wt cardiac and psoas muscle fibers (*Figure 5—figure supplement 2*). Thus, we prepared permeabilized Hom TC cardiac trabeculae muscle segments as we did for the skeletal fibers, stretched them to an SL of 2.2–2.3 μm (high end of SL range in mouse cardiac muscle), and labeled them with a central I-band titin (cardiac N2B element) antibody (*Figure 5C*). Untreated fibers showed normal distribution and order of N2B antibody labels. As a control, we treated permeabilized cardiac fibers with the $Ca^{2+}$-independent gelsolin fragment and confirmed by transmission/immuno-EM that titins are structurally intact after thin-filament removal (*Figure 5—figure supplement 3*). Permeabilized cardiomyocytes were then treated with TEV protease and stained against N2B titin, or we first treated the samples with the $Ca^{2+}$-independent gelsolin fragment to remove the thin filaments, and then cleaved titin with TEV protease, before labeling with N2B antibody to observe titin recoil (*Figure 5C*). Strikingly, while TEV-only treated cardiomyocytes demonstrated partial titin recoil, as in skeletal fibers, the removal of actin filaments allowed cleaved titins to recoil all the way to the Z-disk (*Figure 5D*). We conclude that titin can inherently recoil to the Z-disk, but that in many molecules, titin–thin filament interactions inhibit this action.

## Titin cleavage reduces active force and separates thick filaments

We then set out to explore changes to fibers during active isometric contractions, before and after titin cleavage. Permeabilized fibers were activated in a high-$Ca^{2+}$ solution (pCa 5) at 2.6 μm SL before and after TEV-protease treatment (*Figure 6A,B*). With or without TEV treatment, force

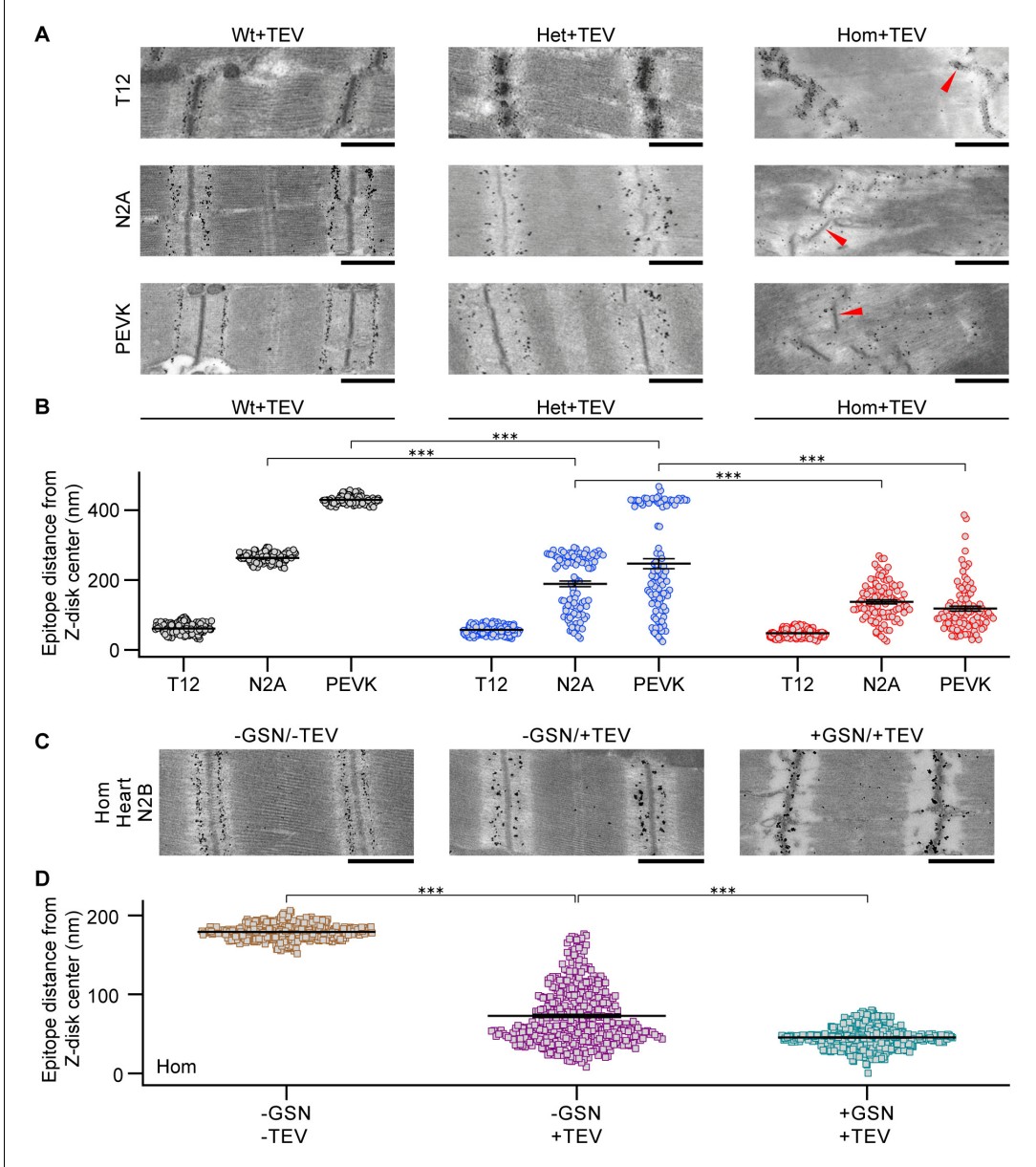

**Figure 5.** Incomplete elastic recoil of cleaved titin to the Z-disk. (**A**) Nanogold immunoelectron micrographs of skeletal fibers labeled with different antibodies to I-band titin (T12, N2A, and proline-glutamate-valine-lysine-rich region (PEVK)). Shown are examples of Wt, Het, and Hom fibers after TEV protease treatment, held passively for 30 min at a stretched length. Red arrows point to Z-disks. (**B**) Recoil of elastic titin to the Z-disk in skeletal fibers quantified by measuring the antibody-epitope to Z-disk-center distance (n = 100 measurements/condition; SL range, 2.9–3.0 μm). (**C**) Passively fixed Hom TC cardiomyocytes labeled for the N2B titin element (central I-band) showing regular staining in controls (−GSN/−TEV; left). Titin cleavage by TEV protease (−GSN/+TEV, middle) or actin-removal by gelsolin treatment, followed by TEV treatment (+GSN/+TEV, right), caused partial (middle) or full (right) recoil of titin springs to Z-disks. (**D**) Quantification of titin recoil in Hom TC cardiomyocytes from images as in (**C**); measurement as in (**B**) (n = 200 measurements/group, evenly from 10 sarcomeres/group; SL range, 2.2–2.3 μm). Stats: ANOVAs with Tukey's HSD post hoc procedure, further confirmed via ranked sum assessment (analysis not shown). Note: error bars are very small.

The online version of this article includes the following source data and figure supplement(s) for figure 5:

**Source data 1.** Data and Stats for *Figure 5*.

**Figure supplement 1.** Immunofluorescence (IF) micrographs of skeletal fibers labeled with different antibodies to I-band titin (T12, N2A, and PEVK, red IF staining), and Z-disk marker α-actinin (ACTN2, green IF staining).

**Figure supplement 2.** Coomassie-stained protein gel of skeletal (psoas) and cardiac muscle tissue.

*Figure 5 continued on next page*

*Figure 5 continued*

**Figure supplement 3.** Representative images of passively fixed, gelsolin-treated but not TEV-treated (+GSN/−TEV), Hom TC cardiomyocytes examined by transmission electron microscopy (EM) (left) or immuno-EM using antibodies to the N2B element of cardiac titin and nanogold-conjugated secondary antibodies (right).

increased after activation to a steady-state level (**Figure 6A**). Without TEV treatment, the first and second contractions produced similar stresses, regardless of genotype. However, TEV treatment after the first contraction significantly reduced active force of the second contraction by 22.83 ±

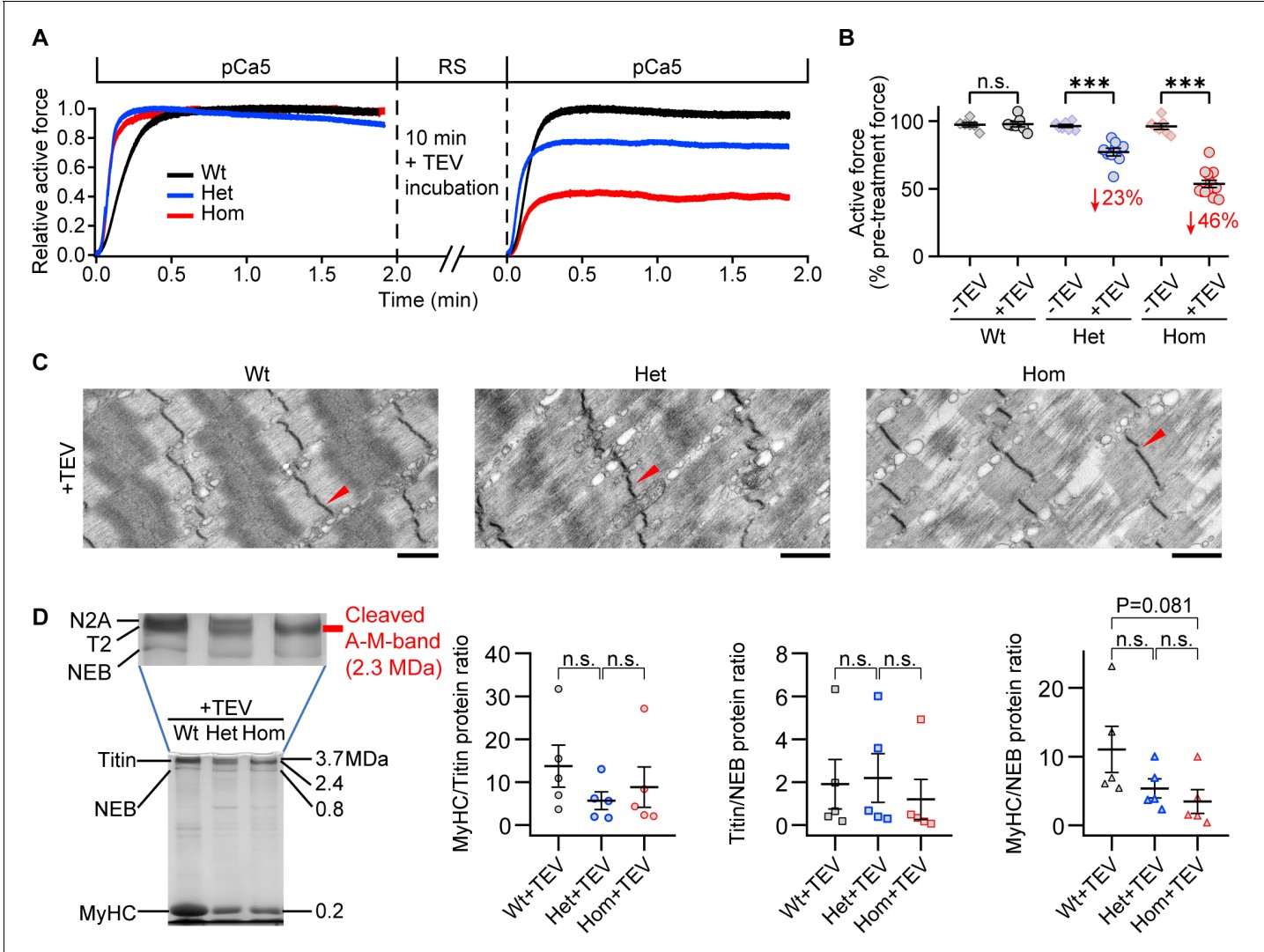

**Figure 6.** Effect of titin cleavage on actively contracting fibers. (**A**) Representative active force traces before and after TEV-protease treatment for Wt (black), Het (blue), and Hom (red) psoas skinned fibers. Each sample was activated once without treatment and once after TEV treatment (SL, 2.6 μm). (**B**) Pooled force data after 2 min of activation at pCa five before/after TEV-protease treatment, normalized to the steady-state active force before TEV treatment (n = 6 Wt-TEV, 7 Wt+TEV, 7 Het-TEV, 9 Het+TEV, 7 Hom-TEV, and 13 Hom+TEV fibers). The average % active force decrease is indicated in red. (**C**) Representative electron micrographs of TEV-treated fibers after active contraction and stretched in the relaxed state. Arrows denote Z-disks. (**D**) A subset of samples were loaded onto Coomassie-stained, loose gels to measure the myosin heavy chain:titin (MyHC/titin), titin/nebulin (titin/NEB), and MyHC/NEB protein ratios after TEV treatment and active contraction. A representative gel (left) and a summary of results (right) are shown (n = 5 fibers/group). Stats: ANOVA with Tukey's HSD post hoc procedure.

The online version of this article includes the following source data for figure 6:

**Source data 1.** Data, Stats and uncropped gel figure for *Figure 6*.

2.78% and 46.31 ± 2.68% in Het and Hom fibers, respectively (*Figure 6B*). Thus, titin-based stiffness is crucial to maintaining high active force levels during muscle contraction.

To also study myocyte ultrastructure, a subset of fibers were prepared for EM after the activation protocol. Het and Hom fibers showed progressive disorganization of Z-disks and A-bands after contraction with 50% and 100% cleaved titins, respectively (*Figure 6C*). The A-band disruption was striking because the damage arose within individual A-bands, among adjacent thick filaments. In normal sarcomeres, the A-band is densely packed with well-organized thick filaments and associated proteins, and so present in EM images as an electron-dense darker 'rectangle' region with a center region of lighter and darker bands that characterizes the M-band, where adjacent thick filaments are tethered together (*Figure 6C*, Wt). However, after titin cleavage, individual A-bands showed a streaming effect of the electron-dense material and a loss of the typical M-band structures (*Figure 6C*), all indicative of a loss of A-band protein order. Remarkably, these images suggested that single or groups of adjacent thick filaments detached from the M-band complex and from one another and traveled in opposite directions until driven against the Z-disks. Furthermore, Hom, and to a lesser extent Het fibers, presented thick filaments that appeared shorter than normal (<1.6 µm), as if the myosin molecules were being shed from the tips of the thick filaments.

To substantiate this observation, we used loose protein gels (2.4% polyacrylamide) to quantify myosin heavy chain (MyHC) protein content relative to titin or nebulin (NEB) protein content (*Figure 6D*). In spite of substantial data scatter, general trends were apparent: the titin/NEB ratio of contracting fibers was largely unaffected by titin cleavage (Wt+TEV: 1.90 ± 1.15; Het+TEV: 2.19 ± 1.13; Hom+TEV: 1.19 ± 0.93), whereas the MyHC/titin ratio tended to be reduced (Wt+TEV: 13.72 ± 4.91; Het+TEV: 5.67 ± 2.05; Hom+TEV: 8.82 ± 4.71), and this trend was even stronger for the MyHC/NEB ratio (Wt+TEV: 11.03 ± 3.36; Het+TEV: 5.37 ± 1.38; Hom+TEV: 3.475 ± 1.73; $p=0.081$, Wt vs. Hom). These findings suggest that myosin molecules are free to float out into solution following titin cleavage and active contraction, emphasizing that contraction with cleaved I-band titin leads to rapid A-band destruction.

To further evaluate whether individual thick filaments travel toward the Z-disk and/or shed thick filament tips, we took a subset of mutant fibers from the activation experiments and labeled the HaloTag with antibodies (*Figure 7A–C*). As the HaloTag localizes at the edges of the thick filaments, we used these markers to assess the relative position and length of individual thick filaments on immunoelectron micrographs. We found that A-band length is smaller after contraction of TEV-treated Het fibers and even shorter for Hom fibers. It is worth noting that the spread of A-band lengths observed in Hom emphasizes that A-band destruction is a dynamic process, of which our analysis only captures a snapshot in time.

As a separate evaluation, we used IF imaging to track both the M-band protein myomesin (MYOM) and the HaloTag in mutant titin. In non-treated (Het) fibers, the MYOM band was cleanly centered between the two HaloTag bands (*Figure 7A*). However, with titin cleavage, MYOM and HaloTag sites began to overlap in Het fibers (*Figure 7B*), and were completely intermingled in Hom fibers, seemingly covering the entire A-band region (*Figure 7C*). Thus, the MYOM-containing M-band appeared to lose its structure and broaden.

These observations present evidence that a lack of titin-based forces leads to (1) the individual thick filaments being pulled out of their regularly aligned arrangement in the A-band and moving toward the Z-disks (an extreme case of A-band 'streaming') and (2) individual myosin molecules becoming detached from the tips of the thick filament (including the D-zone), most likely as a byproduct of the force generated by crossbridge binding to actin. Collectively, our data demonstrate that titin is a critical component to A-band stability during contraction.

Based on the observation of hundreds of EM images, we provide cartoon representations of what we most often visualized in stretched, titin-cleaved, Hom (but also in Het) fibers (*Figure 8*). Compared to controls (*Figure 8A*), after TEV treatment, there is A-band and Z-disk disorder in passively stretched fibers (*Figure 8B*). After the onset of a Ca$^{2+}$-activated fiber-isometric contraction, individual thick filaments begin to separate from the regular alignment of the A-band, as well as decrease in length, suffering from an apparent loss of myosin molecules at the tips of the filament (*Figure 8C*). With prolonged contractions, thick filaments push against (or even translocate through) Z-disks, as suggested by localized densities at some Z-disks (*Figure 8D*). In the most severe cases, sustained contraction (*Figure 8E*) leads to the loss of most structured features in between the Z-disks and generally reduced protein densities, suggesting some proteins (including myosin) have

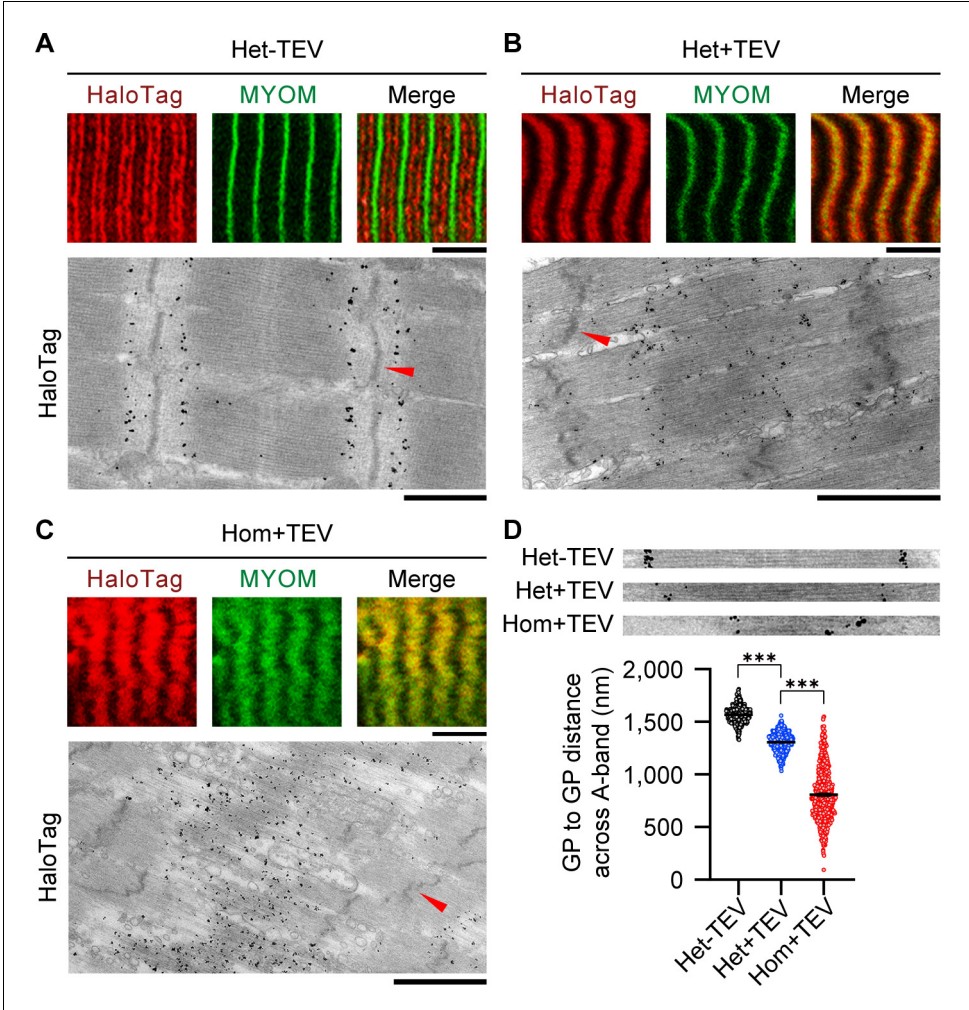

**Figure 7.** Effect of titin cleavage on A-band structure during active contraction. (**A–C**) A-band disorder visualized by tracking the HaloTag position by antibody labeling (red fluorescent secondary antibodies or nanogold particles), using correlative immunofluorescence (IF) and immunoelectron microscopy (IEM). MYOM, anti-myomesin antibody (M-band marker; green). Red arrows indicate Z-disks. Scale bars, 5 μm (IF); 1 μm (IEM). (**D**) HaloTag-to-HaloTag distance after active contraction as a measure of thick filament length, for the three experimental groups, measured on IEM images (n = 50 sarcomeres/group; n = 500 measurements/group; SL, 2.9–3.0 μm). Insets on top show representative examples used for the analysis. Stats: ANOVA with Tukey's HSD post hoc procedure.

The online version of this article includes the following source data for figure 7:

**Source data 1.** Data and Stats for *Figure 7D*.

floated out of the sarcomeres; thick filaments lose their orientation along the long axis of the sarcomere.

## Discussion

This study of titin cleavage in muscles allows for novel insight into as purely a titin-based mechanical effect as possible. From our data, we draw the following main conclusions: (1) Titin-based passive stretch forces in permeabilized fibers scale linearly with the proportion of intact titin springs. (2) Titin contributes at least one-half of the passive viscous and elastic forces of permeabilized fibers in the longitudinal direction. (3) Titin stiffness is critical for maintaining Z-disk organization and structure under mechanical stress. (4) Cleaved titins not only recoil toward the Z-disk elastically but also bind to thin filaments. (5) Titin-based forces are crucial to maintaining high active contractile forces. (6)

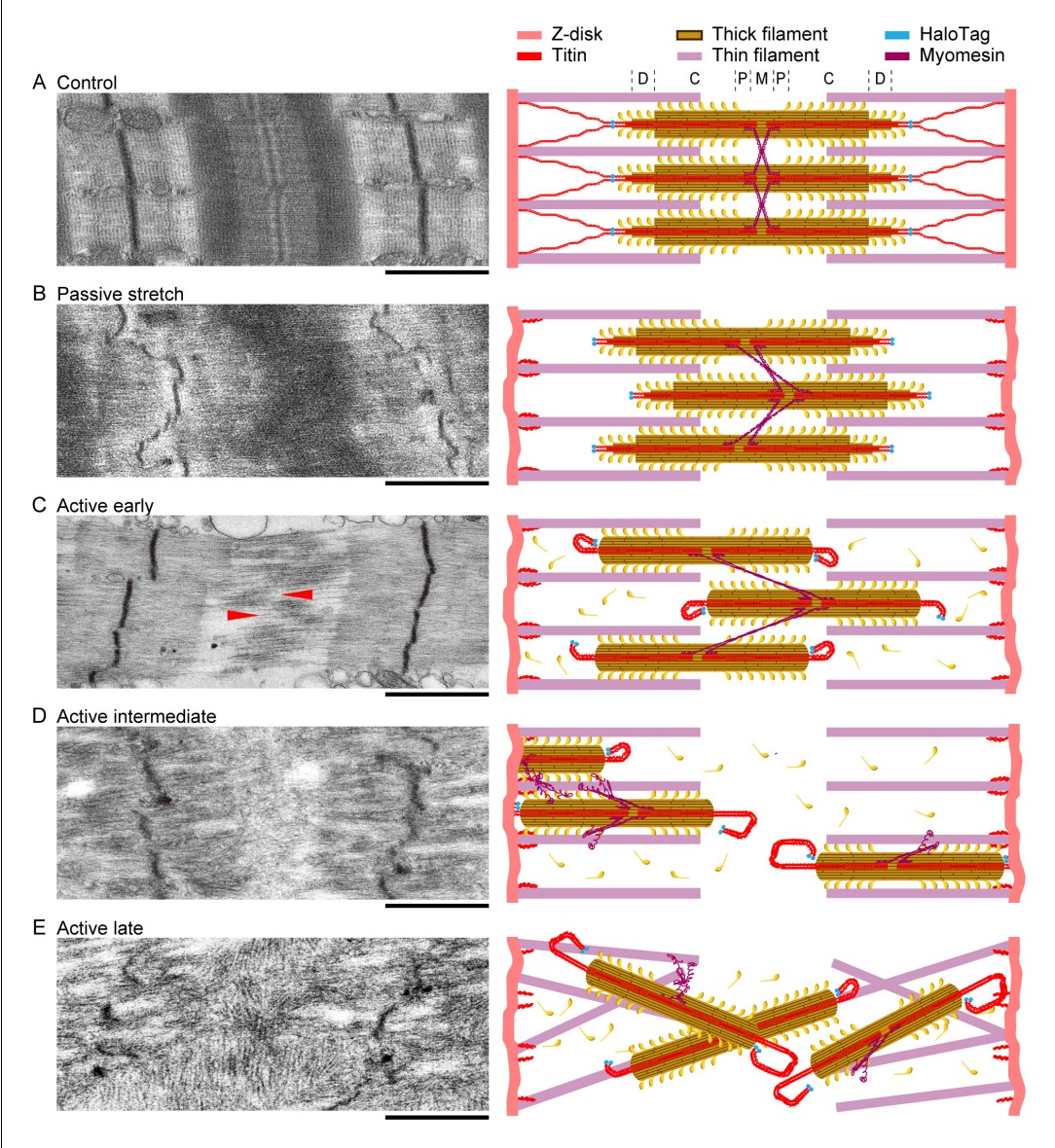

**Figure 8.** Summary of the effects of titin cleavage on sarcomere stability. Representative electron microscopy images (left) and cartoon interpretations (right) of the sarcomere structures for Hom TC permeabilized fibers at a stretched length, as follows: control fibers not treated with TEV protease (**A**), passive fibers treated with TEV protease but never activated (**B**), and activated fibers treated with TEV protease, broken up into early (**C**), intermediate (**D**), and late (**E**) levels of deterioration. Red arrows denote groups of adjacent thick filaments that moved toward opposite Z-disks. Letters D, C, P, and M (top right) refer to the different zones along the thick-filament structure. More details given in Results. Scale bars, 1 μm.

When titin-cleaved Het and Hom TC fibers contract, individual thick filaments quickly begin to separate from each other and travel toward the Z-disks. In parallel, myosin shedding from the thick-filament tips functionally shortens the thick filaments, sometimes up to almost their entire length. Taken together, these findings indicate that titin serves a protective role by preventing active forces from disrupting the A-band organization.

It is well established that the passive force magnitude and the SL at which passive force is measurable are correlated with titin size and stiffness (*Brynnel et al., 2018*; *Mateja et al., 2013*; *Prado et al., 2005*). Furthermore, unspecific titin degradation by trypsin or ionizing radiation in skinned fibers leads to passive force reductions proportional to the amount of titin destroyed (*Higuchi, 1992*; *Horowits et al., 1986*). A linear relationship between the number of intact titin springs and titin-based passive force is clearly supported by our results. If all titins are cleaved in

permeabilized Hom fibers, passive forces drop by a little over 50%, except at the extreme extensions, when other load-bearing elements (e.g. remnant extracellular matrix proteins; *Prado et al., 2005*) become relatively more important.

Surprisingly, in Hom fibers held in a high-stretch state, titin cleavage caused an over-proportionate decrease (~67%) in passive force. The TEV-cleavage site is located in the distal Ig region, where six titin molecules engage in inter-titin interactions for ~100 nm, making up the 'end-filament' (*Liversage et al., 2001*). Beyond the end-filament, in the direction of the Z-disk, titin molecules divide into dimers or monomers (*Kötter et al., 2014*); in the opposite direction, they probably run as dimers along the thick filament (*Zoghbi et al., 2008*). Because of the homotypic interactions within the end filament, it is likely that titin sometimes maintains a connection to its partner titins when cleaved at low passive force and so continues to bear force even when stretched to a long length for a short time (as in our stepwise-stretch protocol). However, in Hom fibers that are held at the long length for an extended time and then TEV treated, all six end-filament titins (per half-thick filament) may retract as a unit, presumably explaining the relatively large drop in passive force. Others have estimated that titin generates >90% of total passive longitudinal force in skinned psoas fibers, when assessed using high-salt extraction baths (*Brynnel et al., 2018*; *Irving et al., 2011*). These high-salt solutions have the disadvantage of also removing other proteins that are potentially load bearing (i.e. intermediate filaments). In contrast, TC fibers allow for target titin cleavage while leaving other proteins intact. Therefore, we provide an update to the current knowledge that in the physiological operating range (<3.4 µm SL), titin contributes less than 70%, but over 50%, of total longitudinal force in permeabilized fibers. It will be interesting to study in future work with the TC mouse model how these values compare to the passive forces of intact fibers or whole muscles.

In contrast to longitudinal, permeabilized fiber transverse stiffness depends to a lesser degree on intact titin, which is responsible for approximately one-third of this stiffness. Other contributors to myocyte stiffness, especially transverse compressive stiffness, may include the microtubules (*Kerr et al., 2015*), the desmin intermediate filaments (*Charrier et al., 2018*), and the non-sarcomeric actin filament (microfilament) network (*Wang et al., 2009*). Glycerol permeabilization procedures on fibers (performed in the current work) degrade some, but not all, of these extramyofibrillar structures, and so their contributions to fiber stiffness will probably be greater for intact fibers or whole muscles (*Brynnel et al., 2018*; *Irving et al., 2011*).

In TEV-treated passive muscle fibers, cleaved titin recoiled toward the Z-disk but also got stuck to thin filaments along the way. Our data are in agreement with the hypotheses that titin functionalities in muscle contraction include titin-actin interactions (*Linke, 2018*; *Nishikawa, 2020*), which have previously been observed in molecular and myofibril preparations (*Dutta et al., 2018*; *Kulke et al., 2001*; *Nagy et al., 2004*). Another interesting implication of our finding is that although titin-based passive forces are frequently not detectable <2.5 µm SL in skeletal muscle (*Linke et al., 1998a*; *Prado et al., 2005*), they must be present even at shorter SLs, as titin was able to recoil all the way to the Z-disk. This makes it clear that titin always has some level of force that can contribute to Z-disk alignment and A-band centering at any SL, at least down to ~1.8 µm SL, where the I-band disappears as thick filaments drive into the Z-disk.

Titin has long been thought to maintain A-band stability during and after active muscle contraction (*Higuchi, 1992*; *Horowits et al., 1986*). *Horowits et al., 1986* used EM and graded radiation-induced titin destruction to demonstrate that titin plays a critical role in centering the A-band complex. Our results fully confirm and extend this 'classic' finding. We show, for the first time, that specific cleavage of the titin springs reduces active forces dramatically and that the tension loss correlates to the amount of titin being cleaved. Strikingly, after activation experiments, the sarcomere structures of TEV-treated Het and Hom fibers present severe disruption and widespread A-band distortion that forcibly detach and deteriorate A-band proteins. Because of their location around a thick filament, the 12 titin molecules (six titins per half-thick filament; *Liversage et al., 2001*) have long been thought to produce a centering force against thick-filament movement (*Horowits, 1999*). These predictions are now demonstrated by the deterioration of thick filaments in titin-cleaved TC fibers.

The question thus arises as to what magnitude of passive elastic forces are generated by titins to sufficiently protect A-band stability during active contraction. The A-band is made up of ~500 parallel thick filaments per µm² cross-sectional area that sit in register and are connected at the M-band through force-bearing M-bridges, predominately made up of myomesin (*Lange et al., 2020*). At

rest, all thick filaments are radially and longitudinally centered in a sarcomere by titin-based forces (*Horowits, 1999*), and so tension on the M-band structures, caused by adjacent thick filaments, is ~0 pN. Importantly, because of thin filament length variability as large as 100 nm (*Littlefield and Fowler, 2002*), the force producing region between each half-thick filament and thin filament is variable at the immediate onset of contraction. Thick-filament movement away from a central position is exacerbated by M-band structures, which can extend up to 70 nm at low force (*Berkemeier et al., 2011*; *Xiao and Gräter, 2014*), allowing thick-filament movement toward the stronger side of the sarcomere during active contraction and enhancing force imbalances especially during prolonged contractions. Considering this within the context of sarcomere geometry, filament overlap, and structural mechanics between thick filaments, we conservatively estimate forces on M-bridges are <50 pN (0–6.25% maximal force at optimal overlap, $P_0$) on average, with pockets that could reach as high as 120 pN (~25% $P_0$) at 2.6–2.7 μm SL. On the other side of the equation, there are the six titin filaments per half-thick filament (*Liversage et al., 2001*), and their forces have been estimated from passive myofibril mechanics (*Linke et al., 1998b*). Thick-filament movement within A-bands will lead to a titin length change of ~35 nm (six lengthening on one side/six shortening on the other side) and generate a titin-based centering force in psoas of ~12 pN/molecule at 2.65 μm SL (*Linke et al., 1998b*; *Prado et al., 2005*). Based on this first approximation, titin-based centering forces are up to an order of magnitude smaller than potential thick-filament movement forces. To produce 120 pN of centering force against our upper estimate of thick filament force, and provide A-band stability, active titin stiffness would need to be approximately 4x larger than measured in passive muscle.

Interestingly, recent hypotheses in muscle contraction purport that titin undergoes a structural rearrangement during contraction that leads to actin binding at or around the N2A/PEVK regions upon activation (*Dutta et al., 2018*; *Kulke et al., 2001*; *Nagy et al., 2004*). Consequently, the functional extensibility of I-band titin is thought to change so that only the stiffer PEVK region, and not the more compliant proximal Ig region, is extensible (*Linke, 2018*; *Nishikawa, 2020*; *Powers et al., 2020*). This structural change has the effect of increasing titin stiffness by fourfold to sixfold (*Nishikawa, 2020*), which is in agreement with our estimation that titin must become up to fourfold stiffer in active, compared to passive, permeabilized fibers. How actin–titin interactions are regulated upon active contraction to increase titin stiffness remains to be addressed in future work.

Finally, an unexpected result was the shedding of myosin proteins off the tips of individual thick filaments during contraction. The atomic structure of the thick filament remains to be fully resolved in mammalian muscles (*Irving, 2017*; *Zoghbi et al., 2008*). However, we do know that the central portion of the thick filament contains an even volume of myosin proteins, whereas the last ~160 nm of the thick-filament ends (including the D-zone; *Figure 8*) taper down until only a few myosin proteins are left, in a potentially unstabilized manner. Because the six titin filaments that run along the half-thick filament all converge at the end of the thick filament into the end-filament (*Houmeida et al., 2008*), it is not difficult to consider that this region of titin can help stabilize the tips of the thick filaments (*Tskhovrebova and Trinick, 2017*). This view is supported by the findings in zebrafish containing an A-band titin null mutation, where prenatal sarcomeres assemble normally, but quickly degrade at the start of prenatal contractile activity, indicating a lack of structural stability and a need for titin proteins for stabilization (*Myhre et al., 2014*). Uncovering the reason for thick-filament instability after titin is cleaved, but not removed from the A-band, will be an interesting future line of investigation.

## Materials and methods

**Key resources table**

| Reagent type (species) or resource | Designation | Source or reference | Identifiers | Additional information |
|---|---|---|---|---|
| Gene *Mus musculus* | Ttn | Ensembl | ENSMUSG00 000051747 | |

*Continued on next page*

*Continued*

| Reagent type (species) or resource | Designation | Source or reference | Identifiers | Additional information |
|---|---|---|---|---|
| Strain background *Mus musculus* | C57BL/6J | *Rivas-Pardo et al., 2020*, UKM animal facility | RRID:IMSR_JAX:000664 | Approved by LANUV NRW, 81–02.04.2019.A472 |
| Genetic reagent *Mus musculus* | Ttn sequence between exons 224 and 234 | doi: 10.1038/s41467-020-15465-9 | | HaloTag-TEV knock-in cassette |
| Antibody | Anti-titin PEVK, rabbit polyclonal | Eurogentec, Belgium | Custom-made | IEM '(1:500)', IF '(1:400)' |
| Antibody | Anti-titin N2A, rabbit polyclonal | Eurogentec, Belgium | Custom-made | IEM '(1:400)', IF '(1:400)' |
| Antibody | Anti-titin T12, mouse monoclonal | Provided by Dr. D.O. Fürst, Bonn, Germany | Custom-made | IEM '(1:100)', IF '(1:100)' |
| Antibody | Anti-titin N2B, rabbit polyclonal | Myomedix, Mannheim, Germany | TTN-3 | IEM '(1:500)' |
| Antibody | Anti-HaloTag, rabbit polyclonal | Promega | Cat. # G928A, RRID:AB_713650 | IEM '(1:100)', IF '(1:100)', WB '(1:1000)' |
| Antibody | Anti-actinin alpha 2, mouse monoclonal | Sigma | Cat. # A7811, RRID:AB_476766 | IF '(1:100)' |
| Antibody | Anti-myomesin-1, mouse monoclonal | Provided by Dr. E. Ehler, London, UK | Custom-made | IF '(1:100)' |
| Antibody | Mouse IgG F(ab')two antibody fluorescein conjugated, goat polyclonal | Rockland | Cat. # 610–1204, RRID:AB_219653 | IF '(1:400)', secondary antibody |
| Antibody | Cythree affinipure goat anti-rabbit IgG, goat polyclonal | Jackson ImmunoResearch | Cat. # 111-165-003, RRID:AB_2338000 | IF '(1:100)", secondary antibody |
| Antibody | Anti-rabbit IgG 1.4 nm nanogold, goat polyclonal | Nanoprobes | Cat. # 2003 | IEM '(1:100)", secondary antibody |
| Recombinant DNA reagent | pMHT238Delta | doi: 10.1016/j.pep.2007.04.013 | | Used to produce TEV protease |
| Sequence-based reagent | Pmin | doi: 10.1038/s41467-020-15465-9 | PCR primers | CGTGGTGGCTTATCTTCTAGC |
| Sequence-based reagent | PRmin | doi: 10.1038/s41467-020-15465-9 | PCR primers | CTGTTGGTTCATGCATCTCC |
| Peptide, recombinant protein | AcTEV protease | ThermoFisher Scientific | Cat. # 12575023 | Used at 10 units $\mu l^{-1}$ |
| Software, algorithm | MATLAB | https://github.com/UKMPhysII/QuantitativeFractureCode | | For quantifying the fracture area of Z-disk lines of sarcomeres |
| Other | Titin | UniProt | A2ASS6-1 | Consensus titin sequence |

## Animal model and muscle preparation

TC mice (*Rivas-Pardo et al., 2020*) were bred and housed at the University Clinic Muenster. Approval from the local authorities was obtained (LANUV NRW, 81–02.04.2019.A472). Genotyping was conducted by PCR. Ear punches were incubated in proteinase K buffer (100 mM Tris–HCl pH 8,

5 mM ethylenediaminetetraacetic acid [EDTA], 0.2% sodium dodecyl sulfate, 200 mM NaCl, 100 µg/ml proteinase K) at 56°C, 650 rpm overnight. The DNA was precipitated with isopropanol and then washed with 70% ethanol. Afterwards the DNA was resuspended in TE buffer (10 mM Tris–HCl, 1 mM EDTA). The following custom primers were used: 5′ cgtggtggcttatcttctagc 3′, 5′ ctgttggttcatgcatctcc 3′. Genetically wild-type (Wt; n = 5 female, 13 male), heterozygous (Het; n = 7 female, 12 male), and homozygous (Hom; n = 10 female, 9 male) adult TC mice (age range, 5 weeks to 1 year) were euthanized by an isoflurane gas overdose and cervical dislocation. Permeabilized fibers were prepared from psoas muscles using standard glycerol-storage techniques (see 'Passive and active force measurements'), and additionally using Triton X-100 (0.5%) as a detergent, when necessary. All procedures were performed according to the guidelines of the local animal care and use committee of the University Clinic Muenster.

## TEV protease production

TEV protease was produced in-house from vector pMHT238Delta (*Rivas-Pardo et al., 2020*) or acquired commercially from ThermoFisher Scientific (AcTEV protease, catalogue no. 12575–023) and used according to the manufacturer's instructions (ThermoFisher Scientific, Dreieich, Germany). In-house expression of TEV protease was induced in BLR (DE3) cells at $OD_{600}$ = 0.6–1.0, using 1 mM IPTG, 3 hr at 37°C. Proteins were purified by Ni-NTA and size-exclusion chromatography and eluted in 10 mM HEPES, pH 7.2, 150 mM NaCl, 1 mM EDTA. Yield was ~8 mg ml$^{-1}$. Purified proteins were divided into small volumes (50 µl each), flash frozen in liquid nitrogen, and stored at −80°C.

To test TEV protease viability, permeabilized Het and Hom TC muscle was digested and analyzed (*Rivas-Pardo et al., 2020*). Briefly, previously extracted skeletal muscle was divided into pieces ~1–2 mm$^3$ and incubated in 100 µl relaxing buffer in the presence of either the commercially bought TEV protease solution (10 units µl$^{-1}$) or the same volume of in-house purified TEV protease in relaxing solution and 0.1 M dithiothreitol (DTT), 1% protease inhibitor cocktail (G6521, Promega, Walldorf, Germany), for up to 6 hr at room temperature (20–22°C). For titin analysis, 1.8% sodium dodecyl sulfate–polyacrylamide gel electrophoresis (SDS–PAGE) was performed as described (*Prado et al., 2005*; *Unger et al., 2017*), and gel bands were quantified by densitometry (e.g. *Figure 1E*) using ImageQuant LAS 4000 Imaging System and ImageQuant TL software (GE Healthcare, Freiburg, Germany). TEV protease viability and quality was assured if the intact N2A titin band (~3.7 MDa) was reduced by ~100%, ~50%, and 0% in treated Hom tissue, Het tissue, and Wt tissue, respectively.

## SDS–PAGE and immunoblotting

To evaluate titin reactivity for quality control and experimental success, fiber samples treated with or without TEV protease were analyzed by agarose-strengthened 1.8–2.4% SDS–PAGE and visualized with Coomassie blue, as previously done (*Prado et al., 2005*; *Unger et al., 2017*). Relative band intensities between intact and cut N2A titin were used to measure % of total titins cut. Western blotting was performed as described elsewhere (*Unger et al., 2017*). To detect HaloTag-TEV titin, we used the primary antibody (anti-HaloTag pAB, Promega 6928) and secondary antibody (anti-rabbit HRP, Acris, Herford, Germany, R1364HRP). Signals from HRP-conjugated secondary antibodies were visualized by chemiluminescence (Amersham ECL start Western blotting detection reagent, GE Healthcare) and recorded using the ImageQuant LAS 4000 Imaging System (GE Healthcare). Signal intensity was quantified by densitometry using the ImageQuant TL software (GE Healthcare). A subset of TEV-treated fibers (n = 5/genotype) were also loaded onto Coomassie-stained, agarose-strengthened, 2.4% SDS–PAGE, titin gels to measure the MyHC:titin, titin:NEB, and MyHC:NEB protein ratios after active contraction.

## Passive and active force measurements

Mechanical experiments were conducted on permeabilized psoas fiber bundles from Wt, Het, and Hom TC mice. Extracted muscles were skinned and stored in a relaxing solution (in mmol l$^{-1}$: potassium propionate [170], magnesium acetate [2.5], MOPS [20], K$_2$EGTA [5], and ATP [2.5], pH 7.0) for 12 hr at 4°C, then in a relaxing:glycerol (50:50) solution, or rigor:glycerol solution (KCl [100], MgCl$_2$ [2], EGTA [5], Tris [10], DTT [1], 50% glycerol, pH 7.0) at −20°C for a minimum of 3 weeks. To limit protein degradation, all solutions contained one tablet of protease inhibitor (Complete, Roche

Diagnostics, Mannheim, Germany) per 100 ml of solution. For activation experiments, fiber activation (high calcium) solutions were prepared (in mmol l$^{-1}$: potassium propionate [170], magnesium acetate [2.5], MOPS [10], and ATP [2.5]; CaEGTA and K$_2$EGTA were mixed at different proportions to obtain the target value of pCa 5 [−log(Ca$^{2+}$)], pH 7.0). Data were collected for six treatment conditions: three genotypes (Wt/Het/Hom) at two treatment levels (with and without TEV protease incubation) each. On the day of experiments, permeabilized muscles were removed from the storage solution and vigorously washed in relaxing solution on ice. Small fiber bundles of 5–15 fibers were separated from the muscle and attached lengthwise to a piezomotor on one end and a force transducer on the other end via aluminum clamps (Scientific Instruments, Heidelberg, Germany). All mechanical experiments were performed at room temperature (20–22°C). Force data were recorded at 1000 Hz. Each fiber bundle was initially suspended in a bath of relaxing solution and then readily transferred to other baths as needed. SL was measured by laser diffraction and initially set to slack length (~2.2 μm SL). Length changes were accomplished by manual or computer-driven means. Experiments were conducted at room temperature (20–22°C).

We first performed passive mechanics on Wt, Het, and Hom fibers in relaxing solution based on established protocols (*Prado et al., 2005*). Muscle fibers were mechanically stretched from 2.2 to 3.4 μm SL in six incremental stretch-hold steps (0.2 μm SL at ~0.2 μm SL s$^{-1}$) and returned to slack length. In between each stretch step, the fiber was held isometrically for 10 s to record stress relaxation. Fibers were repeatedly moved through this protocol (rest time in between trials, 10 min) until force traces became consistent between trials (~2–3 times), and the final run was recorded. Then, the sample was treated with TEV protease in relaxing solution for 10 min – found to be sufficient to achieve cleavage of all TC titin – and the stretch-hold protocol was repeated at regular intervals for 30 min. The final run was recorded and forces compared before and after treatment. To account for typical force decreases over multiple stretch-hold protocols, these experiments were repeated again without TEV protease incubation and these data used as a control baseline. Force was normalized to the peak force during the last stretch at 3.4 μm SL before addition of TEV, as a function of titin cleavage.

We then conducted an experiment to time-resolve the action of TEV protease on TC fibers. Titin cleavage removes titin-based passive fiber forces, and so we monitored passive force during TEV protease treatment as a proxy for TEV protease activity. We stretched fiber bundles slowly (<0.5 μm s$^{-1}$) in relaxing solution to 3.3 μm SL and allowed for stress relaxation for 10 min. Next, the samples were treated with TEV protease in relaxing solution for 30 min while recording force. A 30 min hold protocol naturally decreases passive force in untreated fibers (stress relaxation), and so to remove this force decrease, we normalized force data to time-matched control force. Data were normalized to the starting force right before TEV protease treatment. Data from time points 0, 1, 2, 3, 4, 5, 10, 15, 20, 25, and 30 min after the addition of TEV protease (or no addition, for control) were used for statistical analysis.

We also measured isometric active force in fiber bundles before and after TEV protease treatment. Fiber bundles began in relaxing solution at 2.2 μm SL and were then passively stretched (~0.5 μm s$^{-1}$) to an average SL of 2.6 μm, held for at least 1 min to accommodate stress relaxation, and then activated in activation solution (pCa 5) for ~120 s, at which point force was always plateaued to a steady-state force. Fibers were deactivated by two relaxing solution washes and shortened back to 2.2 μm SL. Fiber bundles were then treated with TEV protease in relaxing solution for at least 10 min (based on findings in passive experiments), washed in fresh relaxing solution, and then the activation protocol repeated. Force data were normalized to the maximum steady-state active force before TEV protease treatment for each fiber bundle. To account for typical force decreases over activation protocols, these experiments were repeated without TEV protease incubation and this data used as a control baseline.

## Radial stiffness measurements by AFM nanoindentation

To determine the contribution of elastic titin to radial compressive forces in myocytes, isolated, relaxed psoas fibers were probed by nanoindentation using an atomic force microscope (Nanowizard 3 AFM system, JPK, Berlin, Germany), as described recently (*Swist et al., 2020*). Permeabilized fibers were prepared as mentioned above, extensively washed in relaxing solution, and homogenized at a relatively low speed (T10, IKA). The resulting suspension was centrifuged at 600 × g for 2.5 min at 4°C and washed three times in relaxing solution. Single fibers were moved from this

solution to an experimental fluid chamber. Inside the chamber, single fibers were placed onto a cover slip coated with Cell-Tak glue (DLW354240, Sigma-Aldrich, Hamburg, Germany). The glue ensured that fibers adhered tightly to the cover glass during data collection. During experiments, performed at room temperature (20–22°C), the pre-calibrated AFM cantilever (spring constant $k_C$ = 0.03–0.04 N m$^{-1}$, Novascan Technologies, Boone, IA, USA) affixed with a mounted spherical polystyrene bead (diameter, 10 μm) periodically indented the fiber at 2 μm s$^{-1}$ up to a pre-set indentation force of 3 nN. From these measurements, the indentation distance and elastic modulus (Young's modulus; E) were calculated as explained previously (*Li et al., 2016*; *Swist et al., 2020*). Indentation and Young's modulus data were pooled by genotype (Wt/Hom) and treatment type (with or without TEV protease incubation) for statistical analysis.

## Transmission-EM and immuno-EM

To visualize the ultrastructure of fibers, muscle samples were fixed in 4% paraformaldehyde, 15% picric acid in 100 mM phosphate-buffered saline (PBS), pH 7.4 for at least 24 hr. Samples were cut into longitudinal sections with a VT 1000S Leica vibratome (Mannheim, Germany) and rinsed twice in PBS. To visualize the HaloTag in situ, samples were blocked in 20% normal goat serum (NGS) for 1 hr and incubated with the HaloTag polyclonal anti-rabbit (Promega, 100-fold dilution) in PBS supplemented with 2% NGS overnight at 4°C. A subset of fibers from passive and active experiments were prepared with one or more of the primary antibodies specific for various parts of I-band titin: PEVK (polyclonal IgG, custom-made; Eurogentec, Seraing, Belgium, 1:500 in PBS), N2A (polyclonal IgG, custom-made; Eurogentec, Belgium, 1:400 in PBS), and T12 (monoclonal IgG, kindly provided by DO Fürst, Bonn, 1:100 in PBS) sites. For cardiomyocytes, the primary antibody for the cardiac-specific N2B titin (TTN-3 against central I-band titin; polyclonal IgG, Myomedix, Mannheim, Germany; 1:500 in PBS) was used in place of N2A. The sections were then triple-washed with PBS and incubated with the respective secondary antibody conjugated to 1.4 nm Nanogold (Nanoprobes, Stony Brook, NY) overnight at 4°C under agitation. After extensive washing, all sections were post-fixed in 1% glutaraldehyde for 10 min and after rinsing, they were reacted with HQ Silver kit (Nanoprobes) to increase the apparent gold particle (GP) diameter. After treatment with OsO$_4$, samples were counterstained with uranyl acetate in 70% ethanol, dehydrated, and embedded in Durcupan resin (Fluka, Buchs, Switzerland). Resin blocks were made and ultrathin sections prepared with a Leica Ultracut S (Mannheim, Germany). Sections were adsorbed to glow-discharged Formvar carbon-coated copper grids. Images were taken using a Zeiss LEO 910 electron microscope (Zeiss, Oberkochen, Germany) equipped with a TRS sharpeye CCD Camera and manufacturer's software (Troendle, Moorenweis, Germany).

## Quantification of immunogold particle distribution

On immune-EM images, we measured the nearest distance between the sarcomeric Z-disk center and the epitope positions for several anti-titin antibodies, including the T12 (peripheral Z-disk), N2A or N2B (central I-band), and PEVK (I-band near A-band edge) antibodies, as described previously (*Unger et al., 2017*), following GP-conjugated secondary antibody labeling. In addition, we measured GP abundance per half-sarcomere on another subset of fibers. To quantify GP distribution, micrographs from at least three independent fiber preparations were analyzed. A minimum of 10 different cells and 50 sarcomeres per experimental condition were included in the analysis. As we intended to analyze the location of titin I-band epitopes, we defined a region of interest containing the sarcomeric I-band including Z-disk, ±20 nm, focusing on areas slightly below the surface of the fibers, because the pre-embedding immuno-EM technique provides best results in more superficial tissue regions. Counting of GPs was done in at least five non-adjacent sarcomeres (minimum, 100 GPs/condition). GP labeling that was considered background staining (e.g. cytosolic and/or extrasarcomeric [such as mitochondria] labeling) was excluded from the analysis. Only GPs with a minimum diameter of 8 nm and an appropriate electron density were included in the analysis. Fused (aggregated) GPs were also assessed if individual GPs were recognizable.

Additionally, in activated fiber bundles stained against the HaloTag with primary antibodies and GP-conjugated secondary antibodies, we measured the end-to-end length of thick filaments using the HaloTag positions as markers of the A-band edge. When individual thick filaments were not resolvable in the A-band, we connected epitopes that fell on a line perpendicular to the long axis of

the sarcomere, on either side of the A-band (see *Figure 7D*). Image processing and epitope distance measurements were done with ImageJ 64 bit Java 1.8 (NIH, Bethesda, USA).

## Quantification of Z-disk disorder

We created a new semi-automatic program that quantifies the level of Z-disk disorder on electron micrographs of muscle samples; we named this parameter 'fracture area'. The value was compared among genotypes (Wt/Het/Hom) after TEV treatment of permeabilized fibers in a stretched state (average SL, 3.3 µm; range, 3.0–3.6 µm). The fracture area is a measure of discontinuities in the typically linear shape of Z-disks, the details of which will be submitted to a web-based collection of protocols; currently accessible at https://github.com/UKMPhysII/QuantitativeFractureCode. Briefly, transmission-EM images of fibers were processed with various filters to isolate the Z-disk structures. From these Z-disks, each pixel of a Z-disk line is assigned an x- and y-coordinate, which are then linearly regressed to create a straight line of best fit. The residuals of these are then weighted by the angular difference between the linear regressions for all Z-disks in the x–y coordinate space, averaged, and further processed to output the fracture area (measured in $nm^2$). Increasing values indicate more Z-disk disorder.

## Confocal laser scanning IF microscopy

Skeletal muscle samples were fixed in 4% paraformaldehyde and 15% saturated picric acid in 100 mM PBS overnight at 4°C, dehydrated via ascending ethanol series, and embedded in paraffin. Thin sections (5–7 µm) were cut with an RM 2235 Leica microtome (Mannheim, Germany). Sections were rehydrated, blocked in peroxidase buffer, and a citrate-EGTA antigen recovery protocol was performed. Slides were rinsed with PBS and then blocked with 5% bovine serum albumin including 0.5% Triton X-100 for 60 min. Subsequently, sections were incubated with primary antibodies overnight at 4°C, using one of the following antibodies against (all dilutions in PBS): HaloTag polyclonal anti-rabbit (G928A, Promega, 1:100); ACTN2 monoclonal anti-mouse (EA-53, Sigma–Aldrich; 1:100), T12 monoclonal anti-mouse (kindly provided by D.O. Fürst, Bonn; 1:100), N2A polyclonal anti-rabbit (custom-made by Eurogentec, Seraing, Belgium), PEVK polyclonal anti-rabbit (Myomedix, Mannheim, Germany, 1:400), and myomesin-1 monoclonal anti-mouse (MYOM clone B4; a kind gift from E. Ehler, London; 1:100). Secondary antibodies were Cy3- or FITC-conjugated IgG (Rockland, Limerick, PA; 1:400), which were incubated overnight at 4°C. Stained samples were embedded in Mowiol supplemented with N-propyl-gallate for bleaching protection and analyzed. IF imaging was performed with a Leica SP8 confocal laser scanning microscope equipped with an HC PL Apo CS2 63x NA 1.4 oil immersion objective.

## Statistics

For each response parameter, full factorial ANOVAs with main effects genotype (WT, Het, Hom) and treatment (with or without TEV protease treatment) or Student's t-tests were used. When ANOVA model effects were significant, a post hoc Tukey's Honestly Significant Difference (HSD) all-pairwise comparison analysis was used to test for differences among group means. Alpha values were set at 0.05, and assumptions of normality and homogeneity of variance were evaluated using the Shapiro–Wilk test of normality, Levene's test for equality of variances, and residual analysis. A best Box–Cox transformation was applied to data, if needed, to meet assumptions. When assumptions were not met, we conducted a non-parametric analysis using a Kruskal–Wallis test (non-parametric ANOVA) and Steele-Dwass method (non-parametric Tukey's HSD multicomparison analysis). Data are presented as mean ± standard error mean (SEM) unless otherwise noted. In figures, significant differences are indicated by n.s., not significant at $p>0.05$, $*p<0.05$, $**p<0.01$, $***p<0.001$. The exact value is given for p-values near the significance threshold of 0.05. Statistical analysis was conducted using Microsoft Excel (v11, Microsoft Inc, Seattle, WA) and GraphPad Prism (V7, GraphPad Software, Inc, San Diego, CA).

## Acknowledgements

We thank Dieter Fürst (Bonn) for providing the T12 titin antibody, Elisabeth Ehler (London) for the gift of the MYOM antibody, Marion von Frieling-Salewsky and Marianne Wilhelmi (Münster) for

excellent technical support, Anna Good (Bochum) for critical text and artistic editing, and Julio Fernandez (New York) for providing the impetus for this study.

## Additional information

### Funding

| Funder | Grant reference number | Author |
|---|---|---|
| German Research Foundation | SFB1002-A08 | Wolfgang A Linke |
| IZKF Muenster | Li1/029/20 | Wolfgang A Linke |
| European Research Area Network on Cardiovascular Disease consortium | MINOTAUR | Wolfgang A Linke |
| MedK Program Medical Faculty Muenster | | Jannik Recker<br>Wolfgang A Linke |

The funders had no role in study design, data collection and interpretation, or the decision to submit the work for publication.

### Author contributions

Yong Li, Conceptualization, Formal analysis, Validation, Investigation, Visualization, Methodology, Writing - original draft, Writing - review and editing; Anthony L Hessel, Formal analysis, Validation, Investigation, Visualization, Methodology, Writing - original draft, Writing - review and editing; Andreas Unger, Investigation, Visualization, Methodology; David Ing, Software, Formal analysis, Methodology; Jannik Recker, Franziska Koser, Investigation; Johanna K Freundt, Formal analysis, Investigation; Wolfgang A Linke, Conceptualization, Resources, Data curation, Formal analysis, Supervision, Funding acquisition, Validation, Visualization, Methodology, Project administration, Writing - review and editing

### Author ORCIDs

Yong Li (iD) http://orcid.org/0000-0001-8184-5977
Wolfgang A Linke (iD) https://orcid.org/0000-0003-0801-3773

### Ethics

Animal experimentation: Laboratory mice were bred and housed at the University Clinic Muenster, in strict accordance with, and approval from the local authorities (LANUV NRW, Germany, 81-02.04.2019.A472). All procedures were performed according to the guidelines of the animal care and use committee of the University Clinic Muenster. Adult mice were euthanized by an isoflurane gas overdose and cervical dislocation, as recommended by the local animal care and use committee, and every effort was made to minimize suffering.

### Decision letter and Author response

Decision letter https://doi.org/10.7554/eLife.64107.sa1
Author response https://doi.org/10.7554/eLife.64107.sa2

## Additional files

### Supplementary files

• Transparent reporting form

### Data availability

All data generated or analysed during this study are included in the manuscript and supporting files. Source data files have been provided for Figures 1 to 7.

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
