## [Decision Letter]

**Acceptance summary:**

Your manuscript presents the most compelling ex vivo experiments to date of how titin contributes to the force produced by the sarcomere, while still inside its native environment, the muscle fiber. It provides also key observations relative to the role of titin in maintaining sarcomere integrity under force.

**Decision letter after peer review:**

Thank you for submitting your article "Graded titin cleavage progressively reduces tension and uncovers the source of A-band stability in contracting muscle" for consideration by *eLife*. Your article has been reviewed by Vivek Malhotra as the Senior Editor, a Reviewing Editor, and three reviewers. The following individuals involved in review of your submission have agreed to reveal their identity: Benjamin L Prosser (Reviewer #1); Nuno Miguel Luis (Reviewer #2); Ivan Vikhlyantsev (Reviewer #3).

The reviewers have discussed the reviews with one another and the Reviewing Editor has drafted this decision to help you prepare a revised submission. Most of these points should be easy to address with simple controls and improved quantification of the data.

We would like to draw your attention to changes in our revision policy that we have made in response to COVID-19 (https://elifesciences.org/articles/57162). Specifically, we are asking editors to accept without delay manuscripts, like yours, that they judge can stand as *eLife* papers without additional data, even if they feel that they would make the manuscript stronger.

Summary:

In this study, a recently described titin-cleavage (TC) mouse model was used to study titin's role in generation of both passive tension and active force. A TEV recognition site is inserted between two distal Ig domains close to the I/A bands border region, allowing to very quickly – in the span of few minutes – cut titin in two via TEV protease digestion. This model by itself does not affect mouse development, muscle structure, or performance, but helps monitor titin cleavage at the boundary of I-band region and A-band in sarcomeres. The authors use this titin-cleaved model in muscle explants for force measurement experimentation.

The authors interrogate the contribution of titin to transverse and longitudinal passive mechanics. They find that depending on the amount of destroyed titin molecules, passive and active forces become progressively reduced in Het and Hom permeabilized psoas fiber bundles. They also perform high-quality EM analysis to assess titin recoil and the destruction to the sarcomere (particularly the thick filament) upon active contraction. From these results, the authors provide evidence that titin has a previously unknown function, namely it can support M-band structures.

Essential revisions:

The following concerns should be addressed or discussed before acceptance:

1) Figure 1Е. To what extent are the authors sure that the band of a protein with molecular weight of 2.3МDa is Cronos, the alternative titin isoform, but not T2 or sometimes called T3 titin fragments?

2) Figure 4C. Considering that multiple proteins contribute to Z-disk alignment and that the I-band portion of cleaved titin recoils to the Z-disk, what could account for the dramatic loss of alignment in the z-disks under passive stretch? See also comment below for Figure 6C.

3) Figure 5C/D. What is the effect on the gelsolin treatment alone?

4) Figure 6C. Why are Hom+ z-disks still reasonably aligned, even more so considering the dramatic effect of active contraction overall? How does the z-disk alignment in the active contraction condition compare to the dramatic loss shown in the passive stretch situation?

5) Figure 6 D. The lane denoted as Nebulin could be a titin degradation product. Lane Nebulin should be lower, near 1/3 of the top portion and 2/3 of the bottom portion of the gel. Notably, there are protein bands seen lower, which could be nebulin. Thus, Western blot analysis should be performed to specify the exact location of nebulin in the similar gel.

Also, loss of myosin has some ambiguity. While it seems quite reasonable that this could occur, the authors attempt to quantify this first by looking at the amount of Titin N2A/myosin ratio in protein gels, which produces somewhat ambiguous results. Is the numerator reduced in this ratio after titin cleavage, and could that contribute to variable results? The loss of myosin is an interesting finding (which could also contribute to the drop in active force of course). Perhaps normalizing to another myofibril component (such as a-actinin) could strengthen this finding, or better yet a fractionation assay separating the myofibrillar compartment from a more soluble compartment may show that myosin appears in the soluble fraction after TEV treatment (prior to other myofibrillar elements), which would greatly strengthen this point.

---

## [Author Response]

Essential revisions:The following concerns should be addressed or discussed before acceptance:1) Figure 1Е. To what extent are the authors sure that the band of a protein with molecular weight of 2.3МDa is Cronos, the alternative titin isoform, but not T2 or sometimes called T3 titin fragments?

We are very confident in our identification of Cronos. In a recent publication (Swist et al., 2020), we used a new anti-mouse Cronos antibody that specifically labels the N-terminus of the Cronos isoform, but no other titin isoform. From these studies, we identified that Cronos is also present in titin gels, running slightly below the proteolytic band known as T2. We would like to refer the reviewer(s) to the published article (Swist et al., 2020). An alternative confirmation of the identity of the Cronos band in human cardiac muscle was obtained by us in the following paper, which used wildtype and Cronos-deficient human iPSC-derived cardiomyocytes: Zaunbrecher et al., 2019.

The current manuscript Results refer to the Swist et al., (2020) paper:

“The ~2.3 MDa titin cleavage product was slightly smaller than a doublet running at ~2.4 MDa, which contains proteolytic titin fragment T2 and the alternative titin isoform, Cronos (Swist et al., 2020) (Figure 1E).”

2) Figure 4C. Considering that multiple proteins contribute to Z-disk alignment and that the I-band portion of cleaved titin recoils to the Z-disk, what could account for the dramatic loss of alignment in the z-disks under passive stretch? See also comment below for Figure 6C.

Z-disk alignment is affected directly by the titin-based forces that are pulling at them (perpendicularly from the long axis of the Z-disk) from two directions. We can safely assume that both sides of the Z-disk have hundreds of attached and evenly distributed titins along its whole length, extending to the adjacent thick filaments. In a healthy sarcomere, titin-based forces from both sides of the Z-disk will equal out, and so the Z-disk structure should stay straight. However, as we start to cleave titin with TEV protease, titins are not all severed at the same time. This leads to unbalanced titin-based forces on the Z-disk in localized regions, which will lead to Z-disk misalignment. At short sarcomere lengths (SLs), the titin-based forces are very small, and so small or no misalignments would happen. However, in our study we sever titins at relatively long lengths (up to 3.4 μm SL), where titin-based forces are much larger, and any titin-based imbalances during TEV protease cleavage will produce large force imbalances around the Z-disk and so lead to misalignment. As can be seen from our EM images in Figure 4 and Figure 5, the amount of Z-disk misalignment is variable, and indeed is not homogenous throughout an individual fiber preparation (i.e., some EM sections show more misalignment than others). We wanted to focus on the idea that titin-based forces can have a meaningful impact on Z-disk alignment.

The concluding sentence of the subsection “Titin cleavage causes Z-disk misalignment under stretch” emphasizes these findings:

“Taken together, the observed decrease in Z-disk linearity with progressive titin cleavage demonstrates the importance of titin-based forces in balancing out the I-bands on opposite sides of a Z-disk, which is an interesting, previously unrecognized, property of titin.”

3) Figure 5C/D. What is the effect on the gelsolin treatment alone?

We now include this control (Hom cardiomyocytes treated with gelsolin but not TEV) in Figure 5—figure supplement 3. After gelsolin-only treatment, anti-N2B antibodies label titins within the I-band, indicating that titins remain intact after thin filament extraction. The N2B epitope labeling is a bit more variable than in control cardiomyocytes with no gelsolin or TEV protease treatment in Figure 5, as is expected after actin removal.

4) Figure 6C. Why are Hom+ z-disks still reasonably aligned, even more so considering the dramatic effect of active contraction overall? How does the z-disk alignment in the active contraction condition compare to the dramatic loss shown in the passive stretch situation?

The is an important observation. Our earlier comment for Z-disk misalignment (Figure 5—figure supplement 3) also discussed this point. In passive samples, TEV protease incubation at ~3.4 μm SL leads to Z-disk misalignment because titin-based forces are relatively large and so produce more Z-disk misalignment. Our activation experiments were conducted at the much shorter length of 2.6 μm SL (optimal cross bridge-based forces but consequently, small titin-based forces). In our activation experiments, titin-cleavage occurred at this relatively short SL, and so small Z-disk effects were observed. Active contraction forces, although relatively large, do not seem to affect the Z-disks alignment as much as the A-band structures. We think that this is because active cross bridge-based forces (pulling on thin filaments attached to the Z-disk) between adjacent sarcomeres are still initially similar, even after titin cleavage, and only start to become imbalanced as the A-bands break down over time. We believe that longer activation times (2+ minutes) lead to further sarcomere breakdown, to the point where Z-disks are also highly damaged. Just as in the passive muscle preparations, samples from active experiments show variable sarcomere damage throughout, including very strong Z-disk misalignment, as can be seen in the EM images from Figure 7B, C and Figure 8D, E. In the end, the thick filaments break down upon contraction, suggesting that titin-based forces/structure help with thick filament maintenance and/or stability.

5) Figure 6 D. The lane denoted as Nebulin could be a titin degradation product. Lane Nebulin should be lower, near 1/3 of the top portion and 2/3 of the bottom portion of the gel. Notably, there are protein bands seen lower, which could be nebulin. Thus, Western blot analysis should be performed to specify the exact location of nebulin in the similar gel.

The reason why the nebulin band in Figure 6D is so close to the titin band(s) is that we used an “unusual” 2.4% SDS-PAGE gel, in order to detect both titin (top of gel) and MyHC (bottom of gel) on the same gel lane. When we use a more “typical” 1.8% SDS-PAGE gel, such as that shown in 1E, which resolves the titin species much better but does not display MyHC, we do observe nebulin in the lower portion of the gel. An example is provided in Author response image 1 (right side). The relative positions of these bands on the gel also depend on the gel running time, the electrical settings used, and a few other minor factors. To help visualize the nebulin band in Figure 6D, we have enlarged the upper region of the gel (Author response image 1, left side). We have stated in the Materials and methods that these gels are prepared with different polyacrylamide concentrations. Since we are sure about the identity of the nebulin band on our gels, we have not performed western blotting against nebulin.

**Author response image 1. sa2fig1:** The left gel is from Figure 6D (2.4% SDS-PAGE), the right gel is a more typical titin gel of 1.8% polyacrylamide (all gels are agarose-strengthened), where the nebulin band position is where most people would expect it.

Also, loss of myosin has some ambiguity. While it seems quite reasonable that this could occur, the authors attempt to quantify this first by looking at the amount of Titin N2A/myosin ratio in protein gels, which produces somewhat ambiguous results. Is the numerator reduced in this ratio after titin cleavage, and could that contribute to variable results? The loss of myosin is an interesting finding (which could also contribute to the drop in active force of course). Perhaps normalizing to another myofibril component (such as a-actinin) could strengthen this finding, or better yet a fractionation assay separating the myofibrillar compartment from a more soluble compartment may show that myosin appears in the soluble fraction after TEV treatment (prior to other myofibrillar elements), which would greatly strengthen this point.

We agree that the analysis of MyHC content requires normalization to some stable protein. In this revision, we have attempted this by also normalizing MyHC protein to nebulin, which should be unaffected by the titin cleavage. We find that the MyHC:NEB ratio shows a strong trend to reduction in TEV-treated, calcium-activated, Hom vs. Wt muscle fibers (new graphs in Figure 6D). Both the MyHC:titin and the MyHC:NEB ratios show a substantial mean reduction from WT to Het and Hom, by a factor of two to three, while a more definitive conclusion was prevented by the large data scatter (which we believe is an inherent property of the activation experiments in the absence of intact titins). However, we think that this result provides strong support for the observation on electron micrographs suggesting that individual thick filaments lose myosin molecules when being activated without the support of titin-based forces. Taken together, our data suggest that in titin-cleaved, activated fibers, the A-band dissolves and floats into the buffer solution.